# Unraveling the mechanism of tip-enhanced molecular energy transfer
Colin V. Coane [1,2], Marco Romanelli [1], Giulia Dall'Osto[1], Rosa Di Felice [2,3] ✉ & Stefano Corni [1,3] ✉

Electronic Energy Transfer (EET) between chromophores is fundamental in many natural light-harvesting complexes, serving as a critical step for solar energy funneling in photosynthetic plants and bacteria. The complicated role of the environment in mediating this process in natural architectures has been addressed by recent scanning tunneling microscope experiments involving EET between two molecules supported on a solid substrate. These measurements demonstrated that EET in such conditions has peculiar features, such as a steep dependence on the donor-acceptor distance, reminiscent of a short-range mechanism more than of a Förster-like process. By using state of the art hybrid ab initio/electromagnetic modeling, here we provide a comprehensive theoretical analysis of tip-enhanced EET. In particular, we show that this process can be understood as a complex interplay of electromagnetic-based molecular plasmonic processes, whose result may effectively mimic short range effects. Therefore, the established identification of an exponential decay with Dexter-like effects does not hold for tip-enhanced EET, and accurate electromagnetic modeling is needed to identify the EET mechanism.

Probing and controlling Electronic Energy Transfer (EET) between chromophores in complex environments has drawn increased scientific attention over the last few decades. The EET process is crucial in many natural light-harvesting complexes[1–3] and is a critical step in photosynthesis[4–7], thus making it extremely interesting and potentially useful for developing devices for human use[8–11]. Because photosynthetic complexes operate surrounded by a natural or artificial environment, understanding the role of this environment in their activity is essential, including its influence on EET. This is an arduous task, and in the case of natural architectures, the systems under investigation are so complex that disentangling the various contributions affecting the overall EET efficiency is far from trivial[12–15]. Nonetheless, the process itself primarily relies on the interactions between distinct molecular species, thus making single-molecule experiments vital for dissecting the EET process and eventually shedding light on how it can be controlled. Experiments on self-standing single molecules are limited by their ability to detect optical signals due to diffraction limits and weak molecular luminescence responses[16,17]. Therefore, a promising strategy developed for investigating energy transfer processes is based on utilizing metal-molecule-metal junctions to confine optical signals in a small region, enhancing molecular responses[18–22].

Recent works[23–29] disclose how scanning tunneling microscopy (STM) may be used to effectively probe single-molecule fluorescence by cleverly harnessing tip surface plasmons. This approach leads to strong photoluminescence (tip-enhanced photoluminescence, TEPL) and electroluminescence (STM-induced luminescence, STML) signals of individual molecules placed underneath an atomistic, metallic STM tip, even reaching sub-molecular resolution in certain cases[29]. Experiments of this kind pave the way for real-space tracking of energy transfer between nearby molecules[30,31]. For instance, Cao et al.[30] utilized tip-molecule-substrate junctions to detect the energy flow between different chromophores while accurately controlling their spatial position. Experiments were carried out in ultra-high vacuum at low temperature (4.5 K) and molecules were deposited on a NaCl trilayer placed on top of a silver metallic substrate. By measuring light intensity emitted by donor (D) and acceptor (A) molecules upon selective excitation of the former via tunneling electrons, they could quantitatively probe the energy transfer process. They showed that the efficiency of EET (denoted $RET_{eff}$ in their work[30], for resonance energy transfer) between a palladium-phthalocyanine (PdPc) donor molecule and a free-base phthalocyanine ($H_2Pc$) acceptor molecule exhibits a fast, exponential-like decay trend as a function of the donor-acceptor (D-A) distance. This measured exponential trend cannot be explained in terms of simple dipole-dipole interactions (Förster theory)[32–34], indicating a more thorough theoretical description of the system is needed to decipher experimental data. In addition to the features of the system that are usually employed to explain EET processes such as the spectral overlap, D-A distance, and the orientation between D-A dipole moments[35–37], environmental effects cannot be

[1]Department of Chemical Sciences, University of Padova, via Marzolo 1, Padova, Italy. [2]Department of Physics and Astronomy, University of Southern California, Los Angeles, CA 90089, USA. [3]CNR Institute of Nanoscience, via Campi 213/A, Modena, Italy. ✉e-mail: difelice@usc.edu; stefano.corni@unipd.it

ignored, and in particular the effect of the metallic nanostructure used to scan the system studied must be accounted for to explain the measured decay trend. The scanning tip can induce new decay pathways and modify existing ones, including the energy transfer process itself[38–40]. Careful attention to the nanostructure's physical features and geometry is required, since tuning its plasmonic frequency may either augment or hinder the EET process[41–48].

In the following, we present a comprehensive theoretical framework for a nanostructure-donor-acceptor system, which is able to describe the tip-mediated EET rate along with radiative and non-radiative relaxation processes. This framework allows us to shed light on the experimental results described above. This study builds on the previously developed Polarizable Continuum Model-NanoParticle (PCM-NP) approach and related works[2,49–56], which rely on an ab initio quantum mechanical description of molecules interacting with classically-described metallic nanostructures. The theoretical model introduced here allows us to go beyond the dipole-dipole interaction model by including a thorough description of the molecular electronic structure, utilizing full electron densities and including effects on molecular decay pathways induced by the metallic STM tip. Similar approaches that have been used in this context have shown that a proper ab initio description of target molecules in such sophisticated plasmonic structures is necessary to fully capture subtle plasmon-molecule interactions that can be revealed by experiments targeting sub-molecular resolution[57–61]. With regard to EET in STM junctions, Kong et al.[31] have recently shown that for D-A distances > 1.7 nm, the use of full transition densities instead of the dipole-dipole approximation to evaluate the direct (metal free) D-A EET rate still leads to an $R^{-6}$ Förster-like dependence of the acceptor emission intensity upon donor tunneling excitation, which only approximately matches the experimental trend that is observed. In their modelling, the effect of the plasmonic system in mediating molecular decay pathways (including EET) is not included. Notably, we find here that the effect of the metal on the molecular decay rates in systems coupled to a metallic nanostructure is not negligible over a wide range of intermolecular distances[49,51,62–64]. Quite surprisingly, our calculations show how the proper accounting of all the relevant molecule-metal and molecule-molecule electromagnetic interaction pathways results in a trend that deceptively mimics an exponential decay. In other words, we disclose a situation where the popular criterion to distinguish between Förster-like and Dexter-like energy transfer mechanisms is no longer appropriate due to the relevance of plasmonic nanoscale effects occurring in tip-molecule-substrate STM junctions.

## Results

### Metal-mediated RET efficiency

In the experimental work of Cao et al.[30], the definition of "RET efficiency", $\text{RET}_{\text{eff}}$ is based on emission intensities of the acceptor ($I_A$) and donor molecules ($I_D$) upon excitation of the donor,

$$\text{RET}_{\text{eff}} = \frac{I_A}{I_A + I_D}. \tag{1}$$

However, we highlight that the above quantity $\text{RET}_{\text{eff}}$ does not depend only on the theoretical efficiency of the EET step unless special restrictive conditions are met, such as the donor and acceptor molecules being identical, ideal emitters.

More generally, such empirical energy transfer efficiency depends on multiple radiative and nonradiative decay processes within the system, and the presence of the metallic tip may influence and modify these processes, as schematically illustrated in Fig. 1.

Given the fluorescence quantum yield of the donor $\Phi_D$, its emission intensity reads

$$I_D = \Gamma_{\text{ex}} \cdot \Phi_D = \Gamma_{\text{ex}} \cdot \frac{\Gamma_{\text{rad,D}}}{\Gamma_{\text{EET}} + \Gamma_{\text{rad,D}} + \Gamma_{\text{nr,met,D}} + \Gamma_{\text{nr,0,D}}} \tag{2}$$

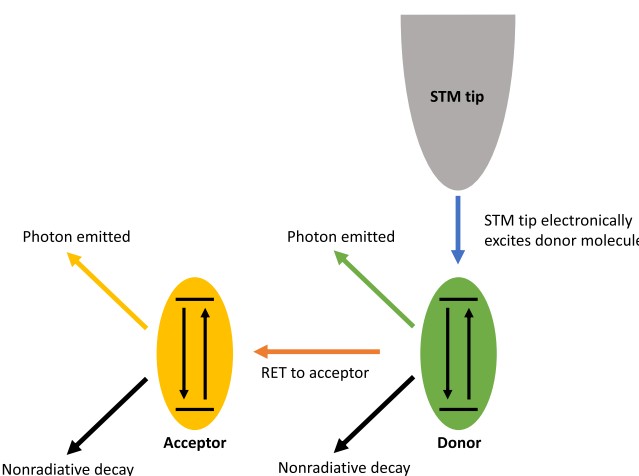

**Fig. 1 | Schematic diagram of energy transfer processes in the STM-donor-acceptor system.** The STM tip excites the donor molecule through a tunneling current, and the donor may decay to its ground state radiatively, nonradiatively, or through EET to the acceptor. If the acceptor is excited through EET, it subsequently may decay to its ground state radiatively or nonradiatively. The excitation energy of the donor's first excited state is larger than that of the acceptor's, which typically prevents energy flow back to the donor. All possible decay processes in both donor and acceptor are affected by the presence of the metallic tip. Indeed, the tip may not only modify the radiative emission of each emitter, but also provides an additional source of nonradiative decay for the molecular excited states.

where $\Gamma_{\text{ex}}$ is the excitation rate (in this case promoted by tunneling electrons), $\Gamma_{\text{rad,D}}$ is the radiative decay rate of the donor in presence of the metal tip, $\Gamma_{\text{nr,met,D}}$ is the nonradiative decay rate induced by the metal tip, $\Gamma_{\text{nr,0,D}}$ is the intrinsic, purely-molecular nonradiative decay rate, and $\Gamma_{\text{EET}}$ is the EET rate from the donor to the acceptor, which is also modified by the presence of the metal tip.

On the other hand, the emission intensity of the acceptor, assuming that the donor undergoes EET to the acceptor upon excitation, is:

$$I_A = \Gamma_{\text{ex}} \cdot \eta_{\text{EET}} \cdot \Phi_A = \Gamma_{\text{ex}} \cdot \frac{\Gamma_{\text{EET}}}{\Gamma_{\text{EET}} + \Gamma_{\text{rad,D}} + \Gamma_{\text{nr,met,D}} + \Gamma_{\text{nr,0,D}}}$$
$$\cdot \frac{\Gamma_{\text{rad,A}}}{\Gamma_{\text{rad,A}} + \Gamma_{\text{nr,met,A}} + \Gamma_{\text{nr,0,A}}}, \tag{3}$$

where $\Gamma_{\text{rad,A}}$, $\Gamma_{\text{nr,0,A}}$ and $\Gamma_{\text{nr,met,A}}$ are the radiative decay rate, intrinsic nonradiative decay rate, and metal-induced nonradiative decay rate of the acceptor, respectively. We note that the quantity $\eta_{\text{EET}}$ is the general theoretical definition of the efficiency of EET from the donor to the acceptor.

Equations (2) and (3) assume only one donor state and one acceptor state, which is a good approximation in the absence of degenerate excited states. However, for the donor molecule (palladium-phthalocyanine, PdPc) the first two excited states are degenerate, so they both may be excited by the tip and participate in the EET process. Moreover, the acceptor molecule (free-base phthalocyanine, $H_2Pc$) has two excited states that are close in energy and may both be excited by EET from the donor. To account for these degeneracies, we consider the emission intensity of the donor from state i, namely

$$I_D^i = \Gamma_{\text{ex}}^i \cdot \Phi_D^i = \Gamma_{\text{ex}}^i \cdot \frac{\Gamma_{\text{rad,D}}^i}{\sum_j \Gamma_{\text{EET}}^{i \to j} + \Gamma_{\text{rad,D}}^i + \Gamma_{\text{nr,met,D}}^i + \Gamma_{\text{nr,0,D}}^i}, \tag{4}$$

where the index i indicates the i-th excited state of the donor, and the index j indicates the j-th excited state of the acceptor.

Likewise, the total acceptor emission intensity after tip-induced excitation of donor state i ($I_A^i$) is the sum over emission intensities from possible

acceptor states j, after EET between donor state i and acceptor state j,

$$I_A^i = \Gamma_{ex}^i \cdot \sum_j \eta_{EET}^{i \to j} \cdot \Phi_A^j, \quad (5)$$

where $\Phi_A^j$ is the jth state emission quantum yield of the acceptor, as defined by the third term in Eq. (3). Thus, the net RET efficiency, as defined in Eq. (1), for a given donor state i becomes

$$(RET_{eff})^i = \frac{\sum_j \eta_{EET}^{i \to j} \cdot \Phi_A^j}{\sum_j \eta_{EET}^{i \to j} \cdot \Phi_A^j + \Phi_D^i} \quad (6)$$

as excitation rates $\Gamma_{ex}^i$ cancel out.

This quantity in Eq. (6) can be evaluated experimentally from the emission intensities considered above. It can also be theoretically obtained from the decay properties of the donor and acceptor molecules alone. Substituting the expressions of $\eta_{EET}^{i \to j}, \Phi_D^i, \Phi_A^j$ into Eq. (6), it is possible to formulate the RET efficiency directly in terms of decay rates,

$$(RET_{eff})^i = \left(1 + \Gamma_{rad,D}^i \cdot \left[\sum_j \frac{\Gamma_{EET}^{i \to j} \cdot \Gamma_{rad,A}^j}{\Gamma_{rad,A}^j + \Gamma_{nr,met,A}^j + \Gamma_{nr,0,A}^j}\right]^{-1}\right)^{-1}. \quad (7)$$

Equation (7) reveals that the RET efficiency upon exciting the i-th donor state is independent of all nonradiative decay properties of the donor, and has the functional form

$$\frac{1}{1+f} \quad (8)$$

where f is a complicated function that depends on the EET rate from donor to acceptor, radiative decay rate of the donor and all decay rates of the acceptor.

Within the PCM-NP framework (see Methods), each of the quantities described in this section can be analytically or numerically determined, retaining a realistic electronic structure description of both donor and acceptor molecules at the ab initio level. The EET rate between molecules in the presence of a metallic nanostructure is given by[46]

$$\Gamma_{EET} = \frac{2\pi}{\hbar} |V_0 + V_{met}|^2 J, \quad (9)$$

where J is the spectral overlap factor, $V_0$ is the electronic coupling between donor and acceptor in vacuum, and $V_{met}$ is the coupling mediated by the metal nanostructure. $V_0$ is calculated as the volume integral over the molecular transition densities of the donor and acceptor $(\rho_X^T)$[65], thus

$$V_0 = \int \rho_A^T(\vec{r}) \rho_D^T(\vec{r}') \frac{1}{|\vec{r} - \vec{r}'|} d\vec{r} d\vec{r}'$$
$$+ \int \rho_A^T(\vec{r}) \rho_D^T(\vec{r}') g_{xc}(\vec{r}, \vec{r}') d\vec{r} d\vec{r}' \quad (10)$$
$$- \omega_0 \int \rho_A^T(\vec{r}) \rho_D^T(\vec{r}) d\vec{r},$$

where $g_{xc}$ is the exchange-correlation kernel and the third term in the right-hand side is the overlap between molecular transition densities weighted by the transition energy. Transition densities for donor emission and acceptor absorption are used instead of dipolar or multipolar approximations to take into account the charge distribution within each molecule during electronic excitations.

$V_{met}$ is expressed in terms of response charges $q_k$ located at the centroid of each k-th tessera on the metal's surface induced by the donor transition

potential[46]. Response charges are multiplied by the acceptor transition potential evaluated at the same spatial coordinates,

$$V_{met} = \sum_{k \in met} \left( \int \rho_A^T(\vec{r}) \frac{1}{|\vec{r} - \vec{s}_k|} d\vec{r} \right) q_k(\vec{s}_k, \varepsilon_{met}(\omega), \rho_D^T) \quad (11)$$

with $\varepsilon_{met}(\omega)$ being the nanostructure's frequency-dependent dielectric function that enters Eq. (17).

As reported in previous work, the radiative decay rate of the donor and acceptor in a generic state b, $\Gamma_{rad,X}^b$ (with X = A, D), in the presence of the metal nanostructure can be evaluated in terms of the sum of the molecular transition dipole in the presence of the metal $\vec{\mu}_{met,X}^b$ and the dipole induced in the nanostructure by the molecular transition density $\vec{\mu}_{ind,X}^b$[53]

$$\Gamma_{rad,X}^b = \frac{4\omega_{b,X}^3}{3\hbar c^3} |\vec{\mu}_{met,X}^b + \vec{\mu}_{ind,X}^b|^2. \quad (12)$$

Additionally, the nonradiative decay rate in the presence of the metal is determined by the imaginary part of the self-interaction between surface response charges and transition potentials evaluated at the same kth tessera[52] on the metal's surface, that is

$$\Gamma_{nr,met,X}^b = -2 \cdot \text{Im}\left\{\sum_k q_k V_k\right\}. \quad (13)$$

On a final note, the intrinsic nonradiative decay rate of the acceptor that enters Eq. (7) ($\Gamma_{nr,0,A}$) is not readily available, but it can be roughly estimated from the vacuum radiative quantum efficiency $\eta_{0,A}$[49]

$$\eta_{0,A} = \frac{\Gamma_{rad,0,A}}{\Gamma_{rad,0,A} + \Gamma_{nr,0,A}}. \quad (14)$$

Since $\Gamma_{rad,0,A}$ can be evaluated considering only the vacuum molecular dipole, $\Gamma_{nr,0,A}$ may be computed if $\eta_{0,A}$ is accessible[49]. Experimental data of radiative quantum efficiency are only available in solution, so for this work we use the solution value $\eta_{0,A} = 0.6$ for $H_2Pc$ as reported in ref. [66]. While $H_2Pc$ is studied here in dry conditions, the same assumption was adopted in a previous work[49] where it was also shown for a similar molecule (ZnPc) that the intrinsic nonradiative decay rate is significantly smaller than other metal-mediated decay rates involved in the process, thus not affecting results whether or not it is included.

## Investigated systems

The chromophores studied here in the presence of an STM tip are a palladium-pthalocyanine (PdPc) donor and a free-base pthalocyanine ($H_2Pc$) acceptor. The two molecules were situated with their aromatic lobes co-planar, lying on the plane that we hereby denote as the xy plane, and treated at the quantum level using Density Functional Theory (DFT) and Time-Dependent Density Functional Theory (TDDFT) when excited state properties are needed. Within the xy plane, the aromatic lobes of each molecule are rotated by 30 degrees from the x and y axes, roughly mimicking the orientation shown by previous experimental STM images[30]. Different D-A distances in the range 1.59–3.20 nm have been investigated to characterize the effect of molecular separation on RET (Fig. 2). Further information related to ab initio calculations and EET modelling based on Eqs. (1)–(13) can be found in Computational Details.

Regarding the plasmonic system, two different STM setups were considered to model the tip-molecule-substrate nanojunctions, illustrated in Fig. 3. One setup was based on a previous STML experimental study[28]: a silver tip was modeled as a truncated cone with rounded edges, with the molecular species adsorbed on a silver cylindrical substrate (Fig. 3a). In the referenced experiment, a three-layer NaCl spacer was placed as a buffer between the metal substrate and the molecules. This insulating buffer was

omitted in our calculations, as previous studies revealed it contributes minimally to the observed molecular response[29,49]. For this geometry, the tip was located 2.0 nm above the substrate to generate a nanocavity hosting the donor molecule, which itself sat 0.5 nm below the tip and 1.5 nm above the substrate. The second setup considered was used in previous single-molecule TEPL calculations[49] and consists of a much larger STM tip with an atomistic protrusion at its apex (Fig. 3b). For this case, the tip-molecule vertical separation was set to 0.4 nm and molecule-substrate separation to 1.4 nm.

Following previous works[28,67] and exploiting the knowledge that the experimentally applied bias voltage is negative[30], we assume that tunneling excitation is achieved by initial electron withdrawal from the donor HOMO orbital by the STM tip. Subsequent electron injection into the LUMO or LUMO+1 takes place via the substrate with equal probability, because the LUMO and LUMO+1 are degenerate and there is no justifiable reason why the substrate should prefer one over the other (both orbitals are equally diffused over the substrate surface). As a result of this, both $S_1$ (HOMO → LUMO) and $S_2$ (HOMO → LUMO + 1) degenerate excited states of PdPc can become equally populated upon tunneling, and each state can couple to either one of the first two excited states of the acceptor. This translates to summing over the index $i$ in Eq. (6) to obtain the full RET$_{eff}$ reported below,

$$\text{RET}_{\text{eff}} = \frac{\sum_i I_A^i}{\sum_i \left(I_A^i + I_D^i\right)} \qquad (15)$$

Two different tip positions were tested, as shown in Fig. 4a (labeled black dots 1,2), where the tip apex is either placed above the middle of one peripheral aromatic ring or above the center of a nearby molecular orbital lobe, respectively. In Supplementary Fig. 1 we show that the main results discussed hereafter are not sensitive to this change in the tip position.

Since the plasmonic absorption peak of the two metallic tips was not on-resonance with the excitation frequencies of the donor and acceptor (see Fig. 5), additional calculations were performed to study the effect of the nanostructure plasmonic resonance peak's location on RET. In doing so, the response of the metal, i.e. the response charges and related quantities of Eqs. (11)–(13), were evaluated at different frequencies to model cases in which the donor and acceptor energies were detuned with respect to the plasmonic peak frequency. To do so, donor and acceptor excitation frequencies were shifted by a constant value, keeping the difference between the two the same, $\omega_{DA} = \omega_D - \omega_A \approx 2.11 - 2.02 = 0.09$ eV. This forced frequency shift is illustrated in Fig. 5 (see vertical colored lines), and was done for various frequencies such that: I. the donor frequency matched the tip's plasmonic peak ($\omega_D = 3.02$ eV for tip 3a and 2.35 eV for tip 3b), II. the acceptor frequency matched the plasmonic peak ($\omega_A = 3.02$ eV for tip 3a and 2.35 eV for tip 3b), and III. (IV.) the donor and acceptor frequencies were both below (above) the tip's plasmonic response to fully characterize its effect on EET (see Fig. 6). We remark that in doing so, the donor or acceptor frequency entering Eq. (12) is always the proper molecular one obtained by TDDFT calculations. This means the shifting procedure just mimics results which would have been obtained using the same molecules (same absorption frequencies) but with a different, shifted metallic response, therefore serving as a proxy for modifying the tip's characteristics.

## Numerical evaluation of RET$_{eff}$ and comparison with experiments
In the quasistatic limit, the absorption cross-section of a given metallic nanostructure is related to the imaginary part of its frequency-dependent polarizability, Im[$\alpha(\omega)$], which can be computed from the dipole induced in the nanostructure upon excitation by an external electric field. In Fig. 5 we

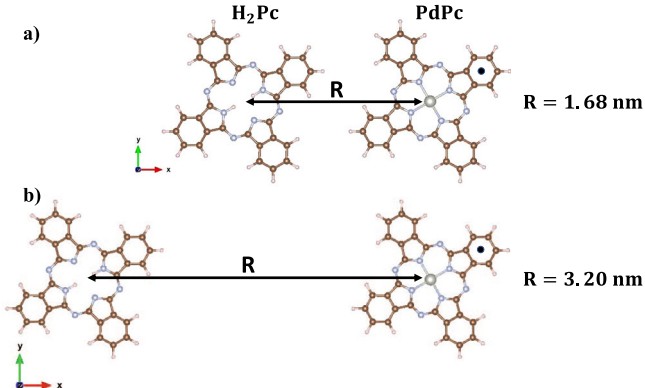

**Fig. 2 | Geometrical configuration of PdPc and H₂Pc molecular structures.** PdPc (donor) and H₂Pc (acceptor) molecules shown at center-center distances of 1.68 nm (**a**) and 3.20 nm (**b**). Aromatic lobes of both molecules are oriented 30 degrees from the x and y axes, as shown. The relative orientation between donor and acceptor is kept rigid throughout the samples separation range.

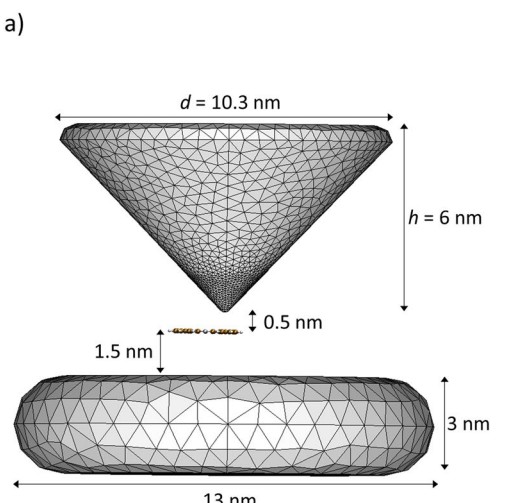

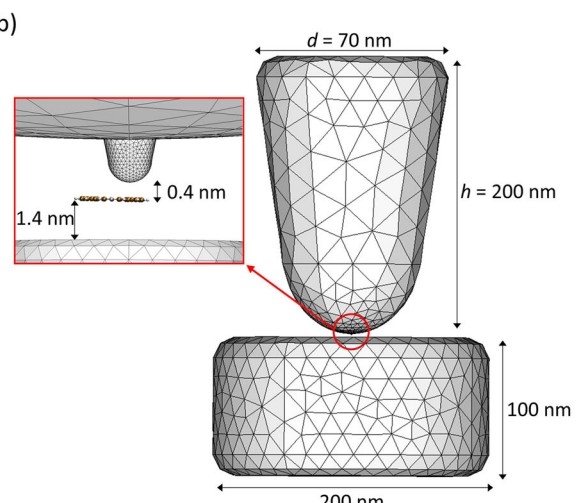

**Fig. 3 | Computational models of the STM structures. a** Nanojunction model employed in EET calculations, based on a previous experimental STML study[28]. Both tip and substrate are made of silver. The tip has a terminal spherical curvature of radius 0.2 nm. **b** Alternative nanojunction model employed in EET calculations, taken from a previous computational TEPL work[49]. Both tip and substrate are made of silver. The tip features an atomistic protrusion (close-up, red box) with a base radius of 0.6 nm and a radius of 0.5 nm for the terminal spherical cap. In both panels **a**, **b** only the PdPc (donor) molecule is shown and it is placed such that the corresponding tip edge is directly above one hexagonal lobe (see also Fig. 2, black dots).

**Fig. 4 | Molecular structures and properties.**
**a** Transition dipoles of the first two excited states of the donor molecule PdPc, overlaid on its atomic structure. The black dots correspond to two different locations of the STM tip apex above the molecule, which have been sampled in this work. **b** Computed HOMO density of both donor and acceptor molecules, displaying overlap between the two.

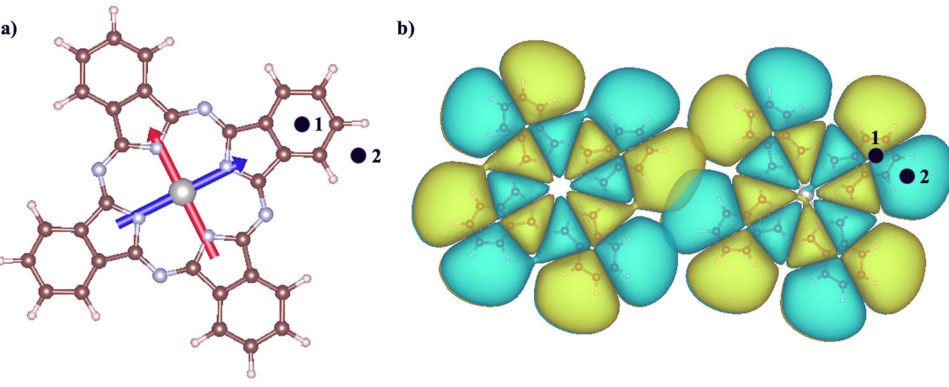

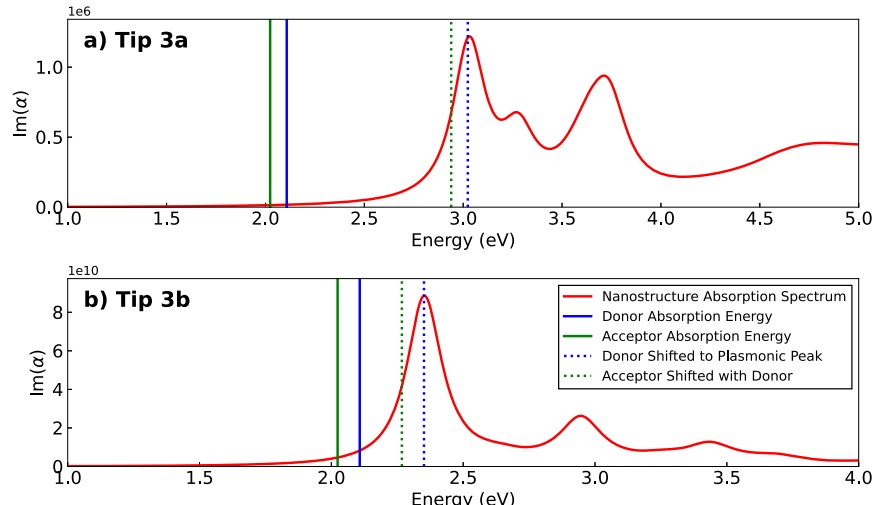

**Fig. 5 | Optical response of the STM structures. a, b** Optical absorption spectra of the STM-like silver tips of Figs. 3**a, b**, computed with the Brendel-Bormann fitting model of the silver dielectric function[71]. The plotted quantity is the imaginary part of the frequency-dependent polarizability $\alpha(\omega)$, which is proportional to the absorption cross-section. Im[$\alpha(\omega)$] was obtained from the electric dipole induced in the corresponding nanostructure upon excitation by an electric field polarized along the tip's z axis (corresponding to the tip central axis). The green (blue) solid line represents the absorption energy of the first excited state of the acceptor (donor). As mentioned previously, the donor's first and second excited states are degenerate, while the acceptor's are close in energy, both of which were considered in EET calculations. Here, only the lowest excited state of the acceptor is plotted for clarity. The green and blue dashed lines represent the excitation energies rigidly shifted so that the donor transition is on resonance with the nanostructure plasmonic peak at 3.02 eV for tip 3a and 2.35 eV for tip 3b. Source data for panels a-b can be found in Supplementary Data 1.

plot the frequency-dependent polarizability for the STM tips of Fig. 3 excited by an electric field polarized along the corresponding tip longitudinal axis (z axis). Absorption energies of the PdPc and $H_2$Pc molecules are overlaid on Fig. 5, and we remark that both donor and acceptor excitation energies (solid green and blue lines) computed with TDDFT are off-resonance with respect to the first bright plasmonic peaks of the nanostructures. However, it was previously shown for similar structures that the plasmonic peak energies are quite sensitive to the choice of the dielectric function model and to detailed geometrical features of the tip which are experimentally unknown[49]. Thus, it is important to assess the impact of the frequency-dependent response of the metal tip on the EET process.

To address this, we performed calculations of relevant frequency-dependent quantities and evaluated the RET efficiency at different excitation frequencies of the donor and acceptor. Such quantities include the frequency-dependent response charges of the tip (Eqs. (16)) and the related metal-affected properties given in Eqs. (11)–(13), which appear in the theoretical expression of the RET efficiency (Eqs. (6), (7) and (15)). Computational results are presented in Fig. 6 for the two tip structures of Fig. 3, along with experimental data from ref. [30]. We find that in all theoretical calculations, RET$_\text{eff}$ decreases monotonically with the distance between molecules, similar to observed trends in the experimental data.

Upon observation, it was found that each curve of computed RET$_\text{eff}$ as a function of distance can be accurately fitted with a simple exponential decay function, RET$_\text{eff}(R) = A_0 e^{-\lambda R}$, as shown in Fig. 6. This effective exponential decay trend is not obvious in the theoretical expression for RET efficiency, but correctly matches trends observed in experiments which could not be explained previously[30]. Indeed, in these experiments it was noted that the observed fast decay of RET with D-A distance could not be understood in terms of simple dipole-dipole interactions (Förster mechanism), and it was speculated that both multipolar RET and Dexter-like energy transfer may be possible explanations for the observed behavior. The finding that a combination of purely electromagnetic effects may mimic an exponential decay with distance is the main result of this present work. The shape of the decay curve is universally used to classify Förster versus Dexter EET mechanisms, and we show here that in case of tip-mediated EET, such a simple indicator cannot be reliably used.

Although the simulated decay of RET$_\text{eff}$ has a somewhat smaller spatial decay parameter $\lambda$ than in experiments, we are able to reproduce the same qualitative exponential trend in terms of bare electrostatic interactions that are combined in a complex manner according to Eq. (6). In fact, the computed exponential trend matches the experimental data when the tip structure of Fig. 3a is used and the donor is close to the resonance condition

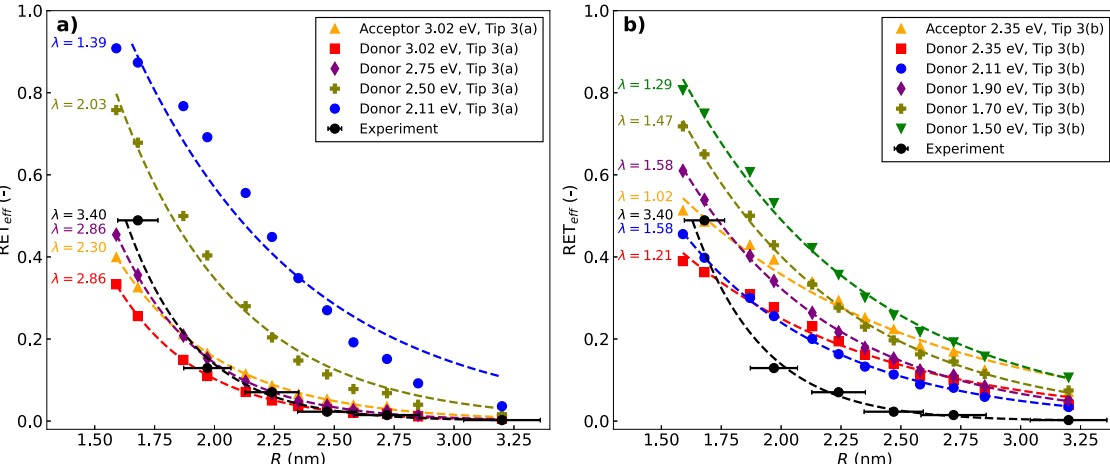

**Fig. 6 | Efficiency of energy transfer. a** RET efficiency as a function of the distance between the centers of donor (PdPc) and acceptor (H$_2$Pc) molecules computed with Eq. (1) (see also Eqs. (2)-(7) and (15)) in the presence of the nanostructure of Fig. 3a. Results are compared with experimental data obtained from ref. [30] (black dots with error bar). The RET$_{eff}$ was evaluated with the donor excitation energy set to: 2.11 eV (unmodified case, dashed blue line), 2.50 eV (dashed olive-green line), 2.75 eV (dashed purple line), and 3.02 eV (i.e., on resonance with the brightest plasmonic peak, dashed red line). Additionally, the RET$_{eff}$ was evaluated with the acceptor excitation energy on-resonance with the brightest plasmonic transition (dashed

yellow line). More details on how this frequency-shift procedure was implemented are given in the Investigated Systems section. Each RET curve has been fitted with an exponential decay function, RET$_{eff}(R) = A_0 e^{-\lambda R}$. For each data set, including experimental data[30], the exponential fitting curve is plotted and the spatial exponential decay constant $\lambda$ is indicated to the left of the corresponding curve. **b** Similar analysis and RET$_{eff}$ curves obtained in the presence of the tip of Fig. 3b. In this case the plasmonic peak resonance is located at 2.35 eV. See also Fig. 5 for a direct comparison of the two tips absorption spectra. Source data for panels **a**, **b** can be found in Supplementary Data 1.

(purple curve of Fig. 6a). This tip structure was effectively used before to model STML experiments on single H$_2$Pc molecules[28].

We note that all the relevant quantities reported in Eqs. (9)–(13) are essentially functions of electromagnetic effects, as the transition density overlap (the last term of Eq. (10)), is completely negligible even at the shortest distance of $\approx$ 1.6 nm. Based on this observation, we disclose that a Dexter-like energy transfer mechanism is of minor relevance here. More precisely, both the exchange and direct density overlap terms of Eq. (10), which are usually related to a Dexter-like process, are 4-5 orders of magnitude smaller than the total coupling value $V_0 + V_{met}$ of Eq. (9), thus pointing out that a Dexter mechanism plays a minor role in the present case. The quantity of interest, RET$_{eff}$, is the result of a rather complex combination of different terms (see Eq. (6), (15)), each having its own frequency and spatial dependence with respect to the metal nanostructure and donor-acceptor distance. Each term is the result of purely classical electromagnetic interactions and should spatially decay as a polynomial function of the donor-acceptor or donor/acceptor-tip distance (see Fig. 7). However, the particular combination of these terms yielding RET$_{eff}$ according to Eqs. (6), (7), (15) can be reasonably well fitted by an exponential curve even if the quantum overlap of the donor and acceptor wavefunctions is negligible.

Looking closely at Fig. 6a, we find a remarkable dependence of the spatial decay rate of the RET$_{eff}$ on the resonance condition between the molecules and the tip. Specifically, moving from a condition in which either the acceptor or donor absorption frequency is exactly on-resonance with the tip's plasmonic peak (yellow and red curve, respectively), to progressively more off-resonance conditions (purple to blue curves), leads to a shallower decay of RET$_{eff}$ as a function of distance. This shallower decay is accompanied, on the other hand, by larger absolute values of RET$_{eff}$ at the shortest distances, with magnitudes larger than experimental observations (black dots in Fig. 6a).

The emergent monotonically decreasing behavior of the RET$_{eff}$ and the dependence of its magnitude on the resonance conditions of absorption frequencies can be further investigated by studying how the contributing decay rates vary with intermolecular distance. As shown in Eq. (7), RET$_{eff}$ depends on the metal-affected radiative decay of the donor and the radiative and nonradiative decay rates of the acceptor. In Fig. 7a, b, we plot the various contributing rates computed at the respective donor and acceptor original frequencies ($\omega_D$ = 2.11 eV/$\omega_A$ = 2.02 eV), corresponding to the blue RET$_{eff}$

curve in Fig. 6a. In Fig. 7c, d the same decay rates computed with the donor absorption frequency shifted to the plasmonic peak of tip 3a ($\omega_D$ = 3.02 eV/ $\omega_A$ = 2.93 eV) are reported. We remark that most quantities associated with the donor are independent of intermolecular distance, since in the computational scheme the donor is fixed in space beneath the metallic tip while only the acceptor is translated to vary D-A distance $R$. These plots illustrate that only the metal-mediated rates significantly affect the magnitude of the RET$_{eff}$. Notably, by moving the donor closer to resonance (panels c and d of Fig. 7), we observe a large increase in both the donor's and the acceptor's plasmon-mediated radiative and nonradiative decay rates, ($\Gamma_{rad,D/A}$, $\Gamma_{nr,met,D/A}$), the latter always dominating. Basically, moving closer to resonance leads to an increase of the magnitude of the denominator of Eqs. (6) and (7) resulting in smaller overall RET$_{eff}$ values than at off-resonance conditions. Overall, results indicate that the magnitude of RET$_{eff}$ decreases as the molecular transition energies approach the nanostructure plasmonic transitions. The theoretical treatment developed in this work allows us to attribute these changes to the fast, metal-enhanced decay channels in the donor and acceptor species. Moreover, we highlight that the quantities in Fig. 7 are plotted on a logarithmic scale but do not display a linear dependence on distance, as one may expect from exponentially decaying terms. Decay rates appear either constant with $R$ or curvilinear, thus corroborating the proper role of bare electromagnetic interactions, which scale polynomially with distance. A similar analysis on decay rates when the tip of Fig. 3b is used is reported in Supplementary Fig. 2, whereas the effect of the tip-molecule distance on RET$_{eff}$ for the same structure is shown in Supplementary Fig. 3.

It is important to note the range of magnitude and decay steepness of the theoretical RET$_{eff}$ in calculations on- and off-resonance at different absorption frequencies. As mentioned in the section Investigated systems, shifting the donor and acceptor absorption frequencies $\omega_D$ and $\omega_A$, while maintaining a consistent STM plasmonic response, is analogous to tuning the STM tip's response, i.e. to modifying its geometry. Indeed, calculations performed for the STM-like nanostructure of Fig. 3b yield qualitatively similar but quantitatively different results (see Fig. 6b) with respect to those performed for the smaller STM-like nanostructure of Fig. 3a. For both setups, the RET$_{eff}$ decays exponentially with the donor-acceptor distance. However, the decay rate depends on the setup, being significantly larger and in agreement with experimental data for the setup of Fig. 3a under resonance

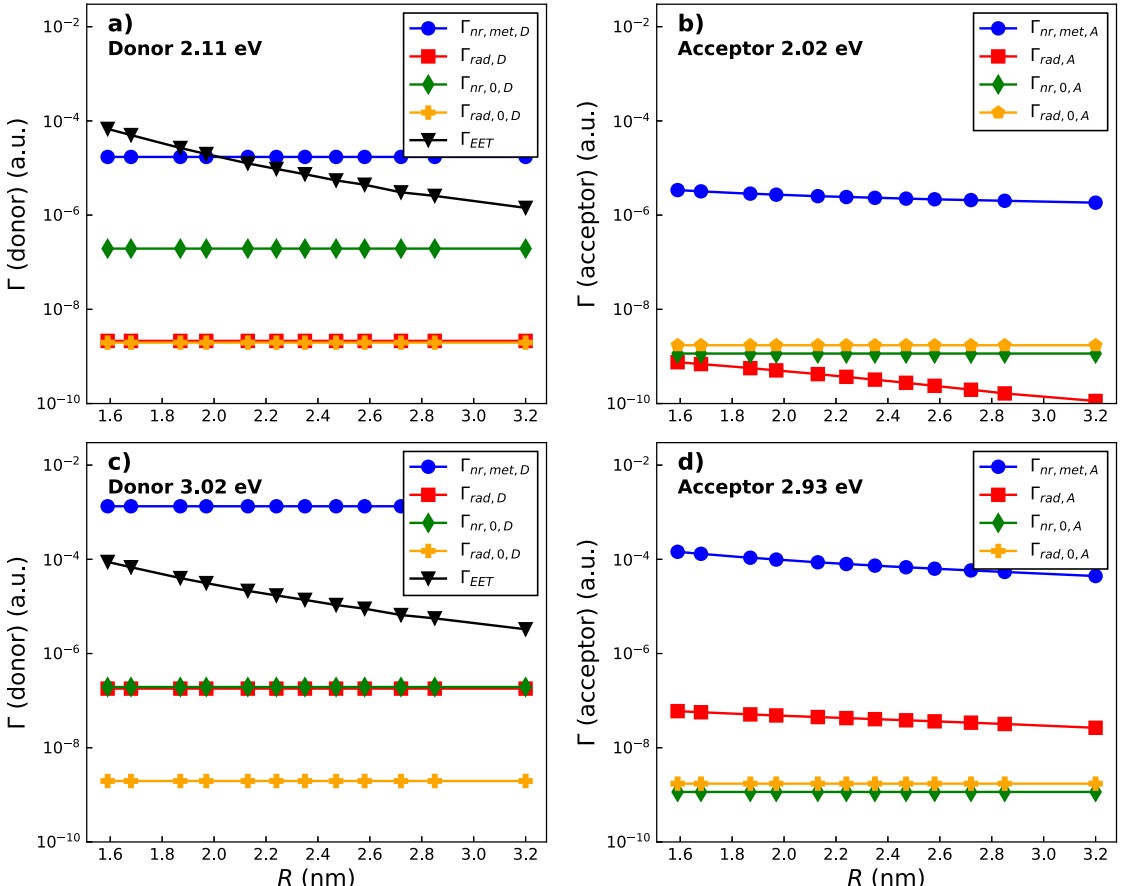

**Fig. 7 | Dependence of individual decay rates on donor-acceptor distance.**
Comparison of donor and acceptor $S_1$ states decay rates that contribute to the EET efficiency as a function of donor-acceptor distance in a logarithmic scale in the presence of the tip structure of Fig. 3a. The metallic response affecting the different rates has been evaluated at the respective donor and acceptor excitation frequencies ($\omega_D \approx 2.11$ eV and $\omega_A \approx 2.02$ eV, panels **a** and **b**, respectively) and with the donor frequency shifted to the tip's resonance peak energy ($\omega_D = 3.02$ eV) while keeping the same difference between donor and acceptor $\omega_{DA} \approx 0.09$ eV (panels c and d, respectively). Panels **a** and **c** show the nonradiative decay rate of the donor induced by the metal ($\Gamma_{nr,met,D}$, blue line, see Eq. (13)), the radiative decay rate of the donor in

the presence of the metal ($\Gamma_{rad,D}$, red line, see Eq. (12)), the intrinsic nonradiative decay rate of the donor ($\Gamma_{nr,0,D}$, green line, computed according to Eq. (14) but for the donor molecule, with $\eta_{0,D} = 5 \times 10^{-4}$, taken from ref. [66]), the intrinsic (vacuum) radiative decay rate of the donor ($\Gamma_{rad,0,D}$, yellow line, see Eq. (14)) and the metal-mediated electronic energy transfer rate ($\Gamma_{EET}$, black line, see Eq. (9)). Panels **b** and **d** show the corresponding quantities for the acceptor molecule (excluding the EET rate), evaluated at the acceptor frequency. A similar analysis in the case of the tip structure of Fig. 3b is reported in Supplementary Fig. 2. Source data for panels **a**-**d** can be found in Supplementary Data 1.

conditions. We remark that the setup of Fig. 3a was previously used to model STML experiments on single $H_2Pc$ molecules[28], while the setup of Fig. 3b was previously used to model TEPL experiments[29,49].

The absorption spectra of the two STM geometries (Fig. 5) are quite different from each other, not only in terms of plasmonic peak frequencies, but also in terms of absorption magnitude, as there is a difference of ≈4–5 orders of magnitude in the maximum value of the imaginary part of the polarizability. These differences between STM geometries, both in absorption spectra and computed decay quantities, highlight the impact of the STM tip geometry on numerical results and in particular on the exact values of the RET$_{eff}$. Consequently, the detailed features of the tip, which are experimentally unavailable, are a plausible source of the remaining small discrepancy between the outcome of our theoretical calculations and the outcome of experiments[30], thus suggesting that a more detailed experimental characterization of these features would be instrumental. We stress the evidence, emerging from our computational work, that the tip geometry plays a fundamental role in the EET process. In fact, we have shown that theoretical results depend on the shape of the modeled tip and that the results obtained for the setup of Fig. 3a quantitatively agree almost perfectly with experiments. This suggests the possibility to further tune the STM tip geometry and absorption spectrum to either maximize or minimize EET

efficiency between donor and acceptor molecules and to control how steeply EET drops off as the molecules are brought farther apart.

## Conclusion

In this work, we build on the previously developed PCM-NP theory[46,50,52,62,68] aimed to describe the interaction between classical nanoparticles and ab initio molecules, extending the procedure to account for energy transfer processes in multichromophoric systems in the presence of a plasmonic nanoparticle. The proposed modelling strategy was applied to an intriguing case study that was recently explored experimentally[30], where a plasmonic STM tip was used to monitor the energy transfer process in a donor (PdPc, palladium-phthalocyanine)-acceptor ($H_2Pc$, bare phthalocyanine) pair as a function of the donor-acceptor distance. This theoretical approach helped clarify and explain the exponential-like decay trend of RET$_{eff} = \frac{I_A}{I_A + I_D}$, which was previously observed[30]. In fact, the main result of this work is that for tip-mediated EET, an almost exponential decay is obtained as a consequence of the complex interplay of purely electromagnetic effects. Thus, the established identification of an exponential decay with a Dexter-like mechanism does not hold for tip-mediated EET. Remarkably, we also observed that the frequency-dependent response of the plasmonic nanostructure, which is strongly dictated by the nanostructure's geometric shape

and metallic composition, can drastically impact the efficiency of energy transfer in such systems, thus paving the way for engineering these systems to control energy flow at the nanoscale. We believe the theoretical model proposed here may be a valuable starting point for exploring the role of plasmonic nanostructures in tailoring energy flow across molecules in more sophisticated multichromophoric architectures, such as artificial and natural light-harvesting complexes.

## Methods

### PCM NP Model

Following previous work[49] on a similar STM-like setup, the coupling between molecular species and the nanostructured metallic tip is described by the PCM-NP model[50,68] (Polarizable Continuum Model-NanoParticle). In this approach, molecular electronic structure is computed using ab initio methods. The resulting molecular charge densities perturb the metal nanostructure, and the nanostructure's response to the molecule-induced external perturbation is treated classically with the Polarizable Continuum Model (PCM), making use of the frequency-dependent dielectric function of the nanostructure. The PCM electromagnetic problem is numerically solved using the Boundary Element Method (BEM), where the surface of the metallic nanostructure is discretized into elementary areas, called tesserae. Each tessera is associated with a polarization charge $q_i(\omega)$ located in its geometrical center $\overrightarrow{\mathbf{s}}_i$ that describes the interaction between the nanostructure and the potential of a nearby molecule, $V_i(\omega)$. Polarization charges are computed on the nanostructure surface as

$$\mathbf{q}(\omega) = \mathbf{Q}(\omega)\mathbf{V}(\omega), \tag{16}$$

where $\mathbf{Q}(\omega)$ is the PCM response matrix in the frequency domain,

$$\mathbf{Q}(\omega) = -\mathbf{S}^{-1}\left(2\pi\frac{\varepsilon(\omega)+1}{\varepsilon(\omega)-1}\mathbf{I} + \mathbf{D}\mathbf{A}\right)^{-1}(2\pi\mathbf{I} + \mathbf{D}\mathbf{A}). \tag{17}$$

Here, $\mathbf{A}$ is a diagonal matrix whose elements are the tesserae areas, while the matrices $\mathbf{S}$ and $\mathbf{D}$ are representative of Calderons' projectors[50],

$$S_{ij} = \frac{1}{|\overrightarrow{\mathbf{s}}_i - \overrightarrow{\mathbf{s}}_j|} \qquad D_{ij} = \frac{(\overrightarrow{\mathbf{s}}_i - \overrightarrow{\mathbf{s}}_j)\cdot\overrightarrow{\mathbf{n}}_j}{|\overrightarrow{\mathbf{s}}_i - \overrightarrow{\mathbf{s}}_j|^3}, \tag{18}$$

where the vector $\overrightarrow{\mathbf{s}}_j$ is representative of the j-th tessera's position on the nanoparticle surface, and $\overrightarrow{\mathbf{n}}_j$ is the unit vector normal to the j-th tessera, pointing outward from the nanoparticle.

### Computational details

Geometry optimization of molecular structures was performed for gas-phase PdPc and $H_2Pc$ with DFT calculations using the software Gaussian16[69]. More specifically, ground state optimization DFT calculations were performed using the B3LYP functional with the LanL2DZ basis set for palladium (Pd) and the 6-31G(d)**++ basis set for non-metal atoms (C, H, N).

The Gmsh code[70] was used for meshing the surface of the tip and substrate and for generating the discretized tesserae. For both tips, meshes were more refined in the proximity of the tip's apex and of the center of the substrate. The setup in Fig. 3a (b) required the use of 6690 (3818) tesserae. In both setups, the Brendel-Bormann fitting model of the dielectric function of silver[71] was used for characterizing the metal's optical response and non-local metal effects are neglected[72].

Utilizing the optimized ground state geometries of the two molecules, electronic energy transfer (EET) between donor and acceptor was calculated in vacuum at a set distance $R$, using TDDFT at the B3LYP/6-31G(d)**++ level of theory for nonmetals and the B3LYP/LanL2DZ level for palladium, consistent with the level of theory used in structural optimizations. Calculations were again performed with Gaussian16[69] to obtain transition dipoles of the first two excited states of each molecule and the vacuum electronic

couplings between each of these excited states. In this regard we note that increasing the basis set size to aug-cc-pVDZ or def2-TZVP as well as using a range-separated functional such as CAM-B3LYP did not lead to significant differences in the value of $V_0$ (Eq. (10)) whose exchange and density overlap contributions always remain 4-5 orders of magnitude smaller that the purely classical electrostatic term.

The first two bright excited states of each molecule that fall in the spectral region probed experimentally[30] are degenerate or close in energy, so both were considered in the simulations.

In addition to the TDDFT calculations of donor-acceptor complexes in vacuum, TDFFT calculations were also carried out for each self-standing molecule in the nanojunction setup. To this end, we computed the molecular electrostatic potential at the center of each nanostructure tessera (see section PCM NP Model), using the same level of theory as above and a modified version of the GAMESS code[73] which accounts for the presence of the metallic structure (see also Supplementary Note 1). We note that the presence of the plasmonic system polarizes the ground state molecular electron densities and so leads to small shift in the molecular excitation energies. Nevertheless this shift is negligible for the investigated setup, in agreement with previous studies[49]. Moreover, at each D-A distance, $R$, considered, the donor remained in the same position and orientation relative to the nanostructure while the acceptor was translated, so calculations were performed for only one donor configuration, but were repeated at each $R$ for the acceptor (atomic coordinates can be found in Supplementary Data 2). The results of the GAMESS calculations were then used to evaluate the polarization charges on the nanostructure surface (Eq. (16)) in the presence of each molecule with the homemade code TDPlas[74]. Additionally, TDPlas calculations yielded radiative decay rates of each molecule, with and without the metal present, as well as nonradiative decay rates mediated by the metal (Eqs. (12) and (13)). Moreover, by taking the real part of the plasmon-molecule self-interaction[49] (Eq. (13)) the plasmon-induced Lamb shift of excitation energies can be assessed. In the present case the largest computed shift value is $\approx 8$ meV for the donor molecule, making it practically negligible. All these quantities are frequency dependent, and the corresponding metal-mediated rates included in Eq. (7) were evaluated at the TDDFT vertical excitation frequencies of the ground state optimized structures of the donor ($S_1/S_2 = 2.11/2.11$ eV) and acceptor molecules ($S_1/S_2 = 2.02/2.05$ eV), respectively. We note that there is a mismatch of $\approx 0.2$ eV between experimental and simulated excitation energies. This systematic small discrepancy does not affect the overall interpretation of the theoretical results discussed in the present work. The spectral overlap value $J$ entering into $\Gamma_{EET}$ of Eq. (9) is set to its experimental value of 1.4 eV$^{-1}$[30]. However, we note that in ref. [30] this value is the estimated spectral overlap between PdPc and the $S_1$ state of $H_2Pc$ ($Q_x$ band). An experimental estimation of the spectral overlap between PdPc and the $S_2$ state of $H_2Pc$ ($Q_y$ band) is absent. However, all results reported in Figs. 6 and 7 are obtained assuming the same $J$ value for both acceptor states, since the RET$_{eff}$ decay profiles are not qualitatively different if the spectral overlap value for the acceptor $S_2$ state is substantially changed, as shown in Supplementary Fig. 4.

We also note that previous works have shown that $H_2Pc$ can undergo tautomerization under tunneling conditions (current-induced tautomerization[75]), even when the tunneling current is not directly passing through the molecule, thus proving that it is an excited-state reaction process[28]. In the work of Cao et al.[30] which we compare our results with, there is no explicit evidence of tautomerization. It could be possible that due to the intense tunneling currents and long spectra acquisition times an average presence of the two tautomers remains buried in the measured signal[26], and so it becomes undetectable. Regardless, the results presented here would effectively take into account such an issue, since upon tautomerization the $S_2$ state accounted for here would convert to $S_1$ of the tautomer, conserving similar electronic properties.

Furthermore, in ref. [31] Kong et al. observed exciton formation, namely an excitation delocalized over more than one molecule, for a similar D-A pair in an STM junction. In our case, at the shortest D-A distance the largest metal-mediated coupling calculated as $V_0 + V_{met}$ under the resonance

condition is rather small, ≈5 meV, thus making exciton formation irrelevant in the present case. Indeed, coherence and exciton formation are not reported in the experimental work of Cao et al.[30].

## Data availability

Source data of Figs. 5-7 can be found in Supplementary Data 1. Atomic coordinates of molecular structures used for calculations can be found in Supplementary Data 2. The authors declare that the data supporting the findings of this study are available within the paper, its Supplementary Information file, and as Supplementary Data files.

## Code availability

The TDPlas code used to model the plasmonic systems and couple them with molecules is freely available at https://github.com/stefano-corni/WaveT_TDPlas. The post-processing python code used to evaluate $RET_{eff}$ and to make the corresponding figures is available from the authors upon reasonable request.

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

## Acknowledgements

The authors acknowledge financial support from the European Union's Horizon 2020 research and innovation program through the project PROID (Grant Agreement No. 964363). Computational work has been carried out on the C3P (Computational Chemistry Community in Padua) HPC facility of the Department of Chemical Sciences of the University of Padua. G.D. and M.R. acknowledge MIUR "Dipartimenti di Eccellenza" under the project Nanochemistry for energy and Health (NExuS) for funding the Ph.D. grant. R.D.F. and C.V.C. acknowledge the USA Department of Energy Office of Science, Basic Energy Sciences, award DE-SC0019432. R.D.F. and C.V.C. acknowledge the Defense Advanced Research Projects Agency (DARPA), Contract No. HR001122C0063.

## Author contributions

C.V.C. performed all the calculations and prepared the images, supported by M.R. and G.D. C.V.C., M.R. and G.D. equally contributed to the first draft of the manuscript. S.C. and R.D.F. supervised the research, and contributed to the paper revision.

## Competing interests

The authors declare no competing interests.
