## [Peer Review File · Communications Chemistry]

Reviewers' comments:

Reviewer #1 (Remarks to the Author):

In this manuscript, Coane and co-workers use computational chemistry techniques to examine the mechanism of plasmonic tip-enhanced energy transfer between a donor and an acceptor molecule placed in the junction between the tip and substrate. The authors find that the effects of the metal tip play an important role in modifying the molecules' decay rates and energy transfer efficiency as does the inclusion of the molecules' electronic structure evaluated from ab initio calculations. The main message of the work is that electromagnetic effects due to plasmon excitation in the tip can lead to exponential decay and that common "rules" to classify Forster versus Dexter energy transfer mechanisms cannot be simply applied to donor-acceptor systems coupled to plasmonic tips. Overall, I think the conclusions of the manuscript are correct and I recommend its acceptance in Communications Chemistry pending satisfactory consideration of the questions/comments below:

1. The most important physical message of the manuscript -- that simple Forster vs Dexter rules based on distance dependent decay profile is not correct in tip-enhanced donor-acceptor systems -- is buried on page 9. The reader should not be forced to read through many pages of computational chemistry minutiae to find this message. Instead it should be made clear from the beginning by rewriting the intro.

2. I believe that the computationally observed exponential decay is due to electromagnetic effects, which is stated by the authors. However, what is the precise physical explanation of these electromagnetic effects? Are these effects all buried deep inside some massive calculations with no simple way to explain them? Can the authors not use a much simpler model, such as at the level of dipole-dipole interactions, to provide a simple digestible physical understanding of these electromagnetic effects?

3. The manuscript focuses far too much on the details of the computational chemistry methods used in the work, none of which are particularly important. This comes at the expense of deprecating the experiment system being modeled and the experimental results in question. The reader should not have to read the experimental paper to learn these details. Instead all important details and observations of the experiment need to be described in the authors' manuscript. And this information needs to be done before getting into any theory and numerical calculations. I found the intro, results, and metal mediated RET efficiency sections quite difficult to follow since I did not understand or appreciate the details of the experiment being modeled and the open questions that it exposed. My recommendation is that authors restructure the intro to explain the experiment ahead of theory.

4. I did not understand how the donor was placed in its excited emissive state in the calculations. Experimentally, the donor excitation was induced by tunneling electrons from the tip, but how was this excitation process modeled computationally?

5. The authors emphasize that they've used sophisticated computational chemistry techniques going beyond simple dipolar couplings, etc, but was this necessary? Could the same qualitative exponential decay not have been calculated from a much simpler electromagnetic dipole coupling model?

Reviewer #2 (Remarks to the Author):

The manuscript by Coane and coworkers reports on a theoretical analysis of EET between two molecules enhanced by the presence of a scanning tunneling microscope junction. The Authors use their theoretical framework to discuss how the energy transfer efficiency depends on the separation between the molecules, how the plasmonic properties of the tip affect this process, and compare their results to previously reported experiments. The analysis presented in the paper is interesting and sheds more light into the understanding of EET in an STM environment, which will be also relevant for the more complete description of the EET process in complex environments in general. The manuscript is well-written and clear. However, there are some critical issues, in particular related to the agreement of the calculations with the experiments and therefore the validity of the used theoretical approach, that need to be addressed before the manuscript can be recommended for publication. Below, I describe in detail my concerns and comments.

1. Fig. 6 of the main text shows the major result of the paper: the dependence of the EET on the molecular separation compared with the experimental data from ref. 30. What is surprising is that the best agreement between the calculation and experiment is obtained for the curves when the donor is strongly detuned from the plasmonic spectrum (curves labeled as donor 1.70 eV and 1.50 eV). However, in such experiments, the plasmonic response of the tip is usually tuned to the molecular resonances to obtain a good signal-to-noise ratio. This condition is represented by the curves labelled as “resonance”, which on the other hand do not reproduce the high efficiency of the process that is observed in the experiments. This key discrepancy is not discussed in the paper and is a critical issue because it shows that the theoretical model either (i) overestimates the nonradiative decay paths or (ii) some key mechanisms are not included properly, therefore its relevance to describe the experiments from ref. 30 or similar can be questioned. Reproducing the general trend (the fast, nearly exponential decay) with purely electromagnetic effects is interesting but addressing the remaining discrepancies is necessary.
2. The Authors assume a fixed tip-molecule distance, how does the EET depend on that parameter?
3. How does the EET change when the tip addresses the other excited state (represented by the red arrow in Fig. 4a)?
4. Some assumptions regarding the molecules are not entirely correct. In the experiments reported in ref. 30, the excited state of PdPc lies slightly below the second excited state of H2Pc, which should therefore not participate in the EET (as assumed by the model). However, one may include a “degeneracy” of the first excited state due to the tautomerization process, which occurs in such a configuration (see ref. 26). This issue should be discussed in the paper.
5. The model presented in this paper takes the excited state energies from the calculations, whereas the spectral overlap from the experiment. It is better when such parameters are consistent. Furthermore, TDDFT is known to not reproduce the excited state energies correctly and indeed there is around 0.2 eV mismatch between the theory and experiment.
6. In the introduction to the model, the Authors refer essentially only to the works from their group. However, there exist several studies from other groups using similar approaches to address light-matter

interaction between molecules and nanoscale plasmons and as of now the manuscript is not well placed in the context of the relevant literature. Some relevant works include:

<https://doi.org/10.1021/acs.nanolett.7b05297> , <https://doi.org/10.1038/ncomms15225> ,
<https://doi.org/10.1103/PhysRevX.12.011012> , <https://doi.org/10.1038/s41467-019-11315-5> ,
<https://doi.org/10.1021/acs.jpcc.9b10256> and similar. Also, some of the cited experimental works (refs. 28, 29, 31) include theoretical calculations and should be discussed in that context. Ref. 31, in particular, features a theoretical description of EET in an STM environment.

7. Ref. 31 reports on the observation of coherence in EET. Can the model presented in this work reproduce such behavior?

8. As the Authors describe, Eq. 10 includes also the Dexter-like energy transfer. Can the Authors quantitatively extract its contribution at short distances?

Minor comments:

1. In the introduction, one reads "This approach leads to strong photoluminescence signals" – the referred approach (from the previous sentence) involves both photoluminescence and electroluminescence, so this sentence should be rephrased for clarity.

2. Bottom of page 8 – there is a typo ("efficiency").

Reviewer #3 (Remarks to the Author):

The paper entitled "Unraveling the Mechanism of Tip-Enhanced Molecular Energy Transfer" is a theoretical study aiming to reproduce and understand experimental results published in [Cao et al., Nat. Chem., 2021, 13, 766-770]. The authors use a theoretical framework combining classical quasi-static material response of metallic objects (scanning-tunneling-microscope [STM] tip and substrate) with ab-initio modelling of electronic excitations in molecules. In particular, the authors study the distance-dependence of resonant-energy-transfer (RET) efficiency between a pair of phthalocyanine molecules PtPc and H2Pc. The main conclusion of the paper is that the distance dependence obtained using the full ab-initio distribution of molecular transition densities (instead of considering just the point-dipole approximation) can be accurately fitted with an exponential function. This conclusion, combined with additional theoretical analysis leads the authors to the conclusion that an exponential fit is not a good-enough signature of the presence of Dexter energy transfer mechanism in this dimer system.

The authors perform DFT calculations in a vacuum using the B3LYP hybrid functional and a gaussian basis set 6-31G(d)**++ with polarization and diffuse functions. They then evaluate the key ingredient - the exciton-exciton coupling - as a sum of a couple of terms reflecting the direct and metal-mediated interaction. The former includes the Coulomb interaction between the respective excitonic transition densities, the contribution coming from the exchange-correlation kernel, and a correction accounting for the overlap between the two densities (Eq. 10). The latter is a metal-mediated interaction that requires the solution of a quasi-static Poisson's equation in the environment of the tip and the substrate with source terms coming from the ab-initio transition density. The approach has been developed by the authors in their previous work and in this paper it has been applied to a dimer of molecules, instead of a molecular monomer.

The paper is interesting potentially for the broader physical-chemistry community, mostly then for the community studying energy transfer on the nanoscale, and for the light-emission-in-STM community. It is also well written (up to some typos). The theory is mostly described clearly, although further clarifications are needed (see below). The work fits into the context of former work (I believe that more work, particularly theory from other groups, could be cited in this context). The paper is certainly publishable but I think that it requires an extension addressing the points below and potential concerns of my colleagues. That would make the work more complete and conclusive - especially because of its ambition to explain a particular experiment. In this case one has to carefully consider and discuss all potentially relevant physical effects.

My main concern is that in the dimer configuration, the resulting coupling between the molecular excitons - particularly the "direct" vacuum contribution - is going to depend on the overlap of the orbitals of the donor and acceptor molecules and is going to be proportional to the exact exchange term (for concreteness, in the HOMO-LUMO approximation - which the authors correctly do not adopt - it would be $\sim \iint \psi_{\text{HOMO}_D}(r)\psi_{\text{HOMO}_A}(r)\psi_{\text{LUMO}_A}(r')\psi_{\text{LUMO}_D}(r')/|r-r'| d^3r d^3r'$). Clearly, this integral (and any other integral describing e.g. charge transfer) can only be evaluated correctly if the molecular orbitals of the monomers are correctly described at the position of the other molecule (more precisely, in the overlap region), i.e. far away from the nuclei of the respective monomers. But since gaussian basis is considered in the calculation, the asymptotic spatial dependence of the orbitals is actually described incorrectly as decaying with the gaussian tail. It is known that this issue leads to qualitative differences when the orbital density of states $|\psi(r)|^2$ is evaluated at larger distances from the molecule as done e.g. to simulate STM images. I therefore also doubt that the Dexter transfer, and particularly its dependence on the molecule-molecule separation, can be describe correctly in this model using this basis set. Another reason that makes me doubt this approach is the hybrid functional used by the authors which does not implement the exact exchange at larger distances (for that a range-separated functional or the exact HF would be needed). That is, I think that the present approach cannot describe the molecule-molecule distance dependence of an important component of the energy transfer mechanism that may still play a significant role in the experiment of Cao et al. I am also unsure about how the authors distinguish between the "electromagnetic" contribution and the "Dexter" mechanism. The Dexter mechanism is itself an effect of Coulomb interaction, and I believe that it is in principle included in the exchange part of eq. 10 (the authors may correct me and clarify this point). For the above mentioned reasons I think that, in the present form, the paper is not able to conclusively disentangle the mechanisms at play in the experiments of Cao et al. (perhaps this would explain the discrepancy with the experiment observed for small distances in Fig. 6 where the theory shows an incorrect trend showing a much weaker distance dependence than the experiment).

Having said that, it is true that the authors concentrate on the electrostatic interaction between the two molecular excitons. I agree that particularly the metal-mediated component could be described with reasonable accuracy even on the level of theory B3LYP/6-31G(d)**++. This part of the discussion is certainly interesting, including the study of the effect of plasmonic resonance on the energy transfer efficiency. However, I am missing a closer characterisation of the plasmonic properties of the tip-substrate system. What is the effective mode volume of this plasmonic "cavity"? What is the field distribution characterising the mode that is resonant with the excitons? How does it compare to other

work? Also, when inspecting the geometry, I noticed that in Fig. 3b the atomistic protrusion is connected to the main body of the tip abruptly, creating thus a sharp corner (aparent in the zoom). Is this a reasonable geometrical representation? Is the sharp edge between the protrusion and the smoother bigger part of the tip influencing the plasmonic mode volume or perhaps even creating an unrealistic field distribution or resonance in the corner region? How does it influence the results? Similarly, the substrate is modelled as a disk of finite size. Is the result dependent on it's size? I think that at least these points deserve more attention of the authors and should be discussed and displayed in the paper (perhaps at least in the supplement). I think that this discussion is particularly important for this paper as it is targeting (semi) quantitative description of the system and directly compares the theoretical data with a concrete experiment.

I am also wondering about the ability of the model - using a local description of the metal optical response - to properly capture the decay of the excitons into the electron-hole pairs and plasmons of higher k vectors of the substrate. Can the authors estimate how much the non-local effects in the metal could influence the lifetime of the monomer excitons? I believe that this could also rescale the results presented by the authors, although I understand that such a study is likely beyond the scope of the paper. Could the authors discuss this effect and potential limitations of their model in this respect?

Getting into more detail, I would like to ask the authors about the following statement: "Following a previous work [28], the donor state mainly excited by the tip due to tunneling electrons corresponds to the one whose dipole points along the molecule's aromatic lobe beneath the tip, illustrated schematically in Fig. 4a." Are the authors sure that this is a matter of excitation probability? It is the photon emission probability that is usually influenced by the tip position. If the molecule is excited by sequential electron tunneling, it is the density of states of the relevant molecular orbitals at the tip position that determines the excitation probability, not the orientation of the excitonic transition dipoles. And in this case HOMO's DOS is totally symmetrical (assuming that it is HOMO that is involved in the tip-mediated tunneling). LUMO's DOS's symmetry is reduced, but not substantially to preferentially excite just one of the two excitons (so even if LUMO is the orbital involved in tip-mediated tunneling the author's statement is not fully justified). Perhaps the authors should comment on this and probably modify the claim.

Also, in case of H2Pc the authors call the excited states quasi-degenerate. There is a separation of ~ 100 meV between the H2Pc excited states and the excitonic photon emission peaks (commonly denoted as Q_x and Q_y) can be clearly spectrally separated and identified in STM-based experiments. Is it still reasonable to talk about (quasi)degeneracy in this case? How would it modify the results if the states were fully treated as non-degenerate?

Also, it is well known that the molecule of H2Pc features tautomerization when excited. Is this relevant for the present study? Could the authors discuss this point?

The authors implement the spectral overlap factor J (eq. 9) as a constant taken from [30]. Why don't they use the theory tools in hand to estimate it computationally to verify the value?

Related to the spectral overlap, could the authors comment on the effect of energy shifts of the excitonic

peaks induced by the plasmonic environment (plasmonic Lamb shift) and the static Stark shift due to the tip-sample voltage (see [Roslawska et al. PRX 12, 011012 (2022)])? Also, the direct and metal-mediated exciton-exciton coupling between the neighbouring monomers could cause spectral shifts, is this taken into account? The shifts could change the spectral overlap between the donor and the acceptor as a function of the relative position of the molecules with respect to the tip and the molecule-molecule separation. Is this effect potentially important or can it be neglected? Could it be quantified?

I am also wondering about the modification of the ground state when DFT calculations are performed in the presence of the substrate and the tip. Do the authors consider this in their model, or is the presence of metal only modifying the linear-response time-dependent part of the calculation? It would be interesting to see how e.g. the image-charge potential of the metal modifies the electronic structure of the molecules. Is the model of the authors able to describe this? I understand that this point (if not implemented already) is likely beyond the scope of the present paper, I am just asking out of curiosity.

Reviewer #1:

We thank the Reviewer for praising our work and stating that “...Overall, I think the conclusions of the manuscript are correct and I recommend its acceptance in *Communications Chemistry* pending satisfactory consideration of the questions/comments below”. Hereafter, we provide our answers to their comments.

Reviewer’s comment n.1: *“The most important physical message of the manuscript -- that simple Forster vs Dexter rules based on distance dependent decay profile is not correct in tip-enhanced donor-acceptor systems -- is buried on page 9. The reader should not be forced to read through many pages of computational chemistry minutiae to find this message. Instead it should be made clear from the beginning by rewriting the intro”*

Authors’ answer. We agree with the Reviewer that the most important finding of the present work should be stated more clearly from the beginning. Thus, we have modified the manuscript so that the message appears early in the manuscript.

Action taken. We have added some sentences at the end of the introduction (see lines 60-62, marked-up version) highlighting the main finding of the present work, which is then described in more detail on page 9. Furthermore, to enhance readability, we have moved some computational details from the section “Investigated systems” to the new section “Computational Details”, under “Methods”.

Reviewer’s comment n.2: *“I believe that the computationally observed exponential decay is due to electromagnetic effects, which is stated by the authors. However, what is the precise physical explanation of these electromagnetic effects? Are these effects all buried deep inside some massive calculations with no simple way to explain them?”*

Authors’ answer. Eqs. 6-7 represent the theoretically derived expression of RET_{eff} that is used to obtain the results shown in Fig. 6, where the exponential distance dependence shows up. We note that all the excited-state decay processes that are illustrated in Fig.1 enter the RET_{eff} expression in a non-trivial way. The point that we try to make in this work is that even if each metal-mediated process that numerically determines the total value of RET_{eff} is due to solely electromagnetic interactions (so the physical explanation is really plasmon-mediated electromagnetic effects, quantum overlap of donor-acceptor wavefunctions is definitely playing a minor role here), those processes are combined in a complex manner according to Eqs. 6-7, yielding a quantity that can be reasonably well fitted by an exponential decay function even if each elementary process can be quantitatively modeled just by taking into account electrostatic effects.

Action taken. We added some sentences on page 9 (see lines 194-200, marked-up version) emphasizing this aspect.

Can the authors not use a much simpler model, such as at the level of dipole-dipole interactions, to provide a simple digestible physical understanding of these electromagnetic effects?”

Authors' answer. It has been previously shown (see Yang, B. et al. Nat. Photonics 14, 693–699, 2020 - ref. 29) that the use of point-dipole models for this kind of experiments where molecules are rather close to STM tips ($< 1\text{nm}$) would not be able to fully capture all relevant details of experimental data, such as molecular lobes pattern in the corresponding STML/TEPL images and the correct magnitude of the output signal. Based on these findings we believe that such a simplified description would not improve the physical understanding of the observed decay, which would be in both cases (either with point dipoles or full densities) mostly due to electromagnetic effects mediated by the plasmonic system and still combined according to Eqs. 6-7. As a matter of fact, a simple model is not appropriate to explain phenomena in the investigated setup, and the text added on page 9 addresses this issue

Reviewer's comment n.3: *“The manuscript focuses far too much on the details of the computational chemistry methods used in the work, none of which are particularly important. This comes at the expense of deprecating the experiment system being modeled and the experimental results in question. The reader should not have to read the experimental paper to learn these details. Instead all important details and observations of the experiment need to be described in the authors' manuscript. And this information needs to be done before getting into any theory and numerical calculations. I found the intro, results, and metal mediated RET efficiency sections quite difficult to follow since I did not understand or appreciate the details of the experiment being modeled and the open questions that it exposed. My recommendation that is that authors restructure the intro to explain the experiment ahead of theory”*

Authors' answer. We agree with the reviewer that many computational details in the section “Investigated systems” may distract the reader from the results presented later. To ease readability and make the presentation of the investigated systems less dense, we moved some technical aspects to the new section “Computational Details”, under “Methods”. However, we respectfully disagree with the reviewer on the need for reporting all the experimental details, because they are not part of our work and should not be repeated in an original publication. The focus of the present work is on the physical interpretation of previously unexplained experimental findings, through the theoretical methodology that we propose. Hence, the explanation of the methodology and of the computational findings is here intentionally prioritized. Regardless, to be more complete-yet-concise on the experimental description, we revised the description of the experimental systems being modeled in the section “Introduction”.

Action taken. We moved some computational details from “Investigated systems” to the new subsection called “Computational Details”, under “Methods”. We adjusted the experimental description on page 1 (see lines 28-33, marked-up version)

Reviewer's comment n.4: *“I did not understand how the donor was placed in its excited emissive state in the calculations. Experimentally, the donor excitation was induced by tunneling electrons from the tip, but how was this excitation process modeled computationally”*

Authors' answer. We thank the Reviewer for raising this question. The underlying assumption, based on previous findings (Nat. nanotechnology 15, 207–211, 2020 (ref. 28);

Nano letters 19, 2803–2811, 2019 (ref. 67)), is that given the negative bias voltage applied in the experiments, the tip removes one electron from the HOMO donor orbital and subsequent electron injection from the substrate takes place. This electron can be equally injected either in the LUMO or LUMO+1 because these two orbitals are degenerate and there is no sound reason why the substrate should prefer one over the other. Consequently, both donor S1 and S2 degenerate excited states, which are respectively described by HOMO->LUMO and HOMO->LUMO+1 transitions, are equally populated. This implies that the properties of both transitions (e.g. excitation energies, transition densities etc.), determined by the *ab-initio* TDDFT calculations, enter the expression of eq. 6. This is how excitation is accounted for in the theoretical model proposed.

Action taken. We have clarified how the donor is placed in its emissive states on page 5, see lines 144-151 marked-up version.

Reviewer’s comment n.5: *“The authors emphasize that they’ve used sophisticated computational chemistry techniques going beyond simple dipolar couplings, etc, but was this necessary? Could the same qualitative exponential decay not have been calculated from a much simpler electromagnetic dipole coupling model?”*

Authors’ answer. As expressed in the previous reply to comment n.2, previous works have shown that when molecules are very close to atomistically-sharp STM tips (<1nm), full transition densities, instead of simple point-dipole models, should be considered in order to attain a reliable representation of the actual system being probed.

Reviewer #2:

We thank the Reviewer for acknowledging that *“The analysis presented in the paper is interesting and sheds more light into the understanding of EET in an STM environment, which will be also relevant for the more complete description of the EET process in complex environments in general. The manuscript is well-written and clear.”* and for raising important issues related to *“the agreement of the calculations with the experiments and therefore the validity of the used theoretical approach”*, which we hereby address.

Reviewer’s comment n.1: *“Fig. 6 of the main text shows the major result of the paper: the dependence of the EET on the molecular separation compared with the experimental data from ref. 30. What is surprising is that the best agreement between the calculation and experiment is obtained for the curves when the donor is strongly detuned from the plasmonic spectrum (curves labeled as donor 1.70 eV and 1.50 eV). However, in such experiments, the plasmonic response of the tip is usually tuned to the molecular resonances to obtain a good signal-to-noise ratio. This condition is represented by the curves labelled as “resonance”, which on the other hand do not reproduce the high efficiency of the process that is observed in the experiments. This key discrepancy is not discussed in the paper and is a critical issue because it shows that the theoretical model either (i) overestimates the nonradiative decay paths or (ii) some key mechanisms are not included properly, therefore its relevance to describe the experiments from ref. 30 or similar can be questioned. Reproducing the general trend (the fast, nearly exponential decay) with purely electromagnetic effects is interesting but addressing the remaining discrepancies is necessary.”*

Authors' answer. The issue raised by the Reviewer is indeed relevant. After taking into account a missing contribution as suggested by comment n.3 of Reviewer 2 and by comment n.5 of Reviewer 3, we indeed find that experimental results are better reproduced at resonance, as supposed by the Reviewer. We have however to remark that excitation in the targeted experiments is obtained by electron tunneling rather than by photoexcitation, and resonance conditions are likely less stringent than in photoluminescence experiments see, for instance, recent TEPL works: *Nat. Photonics* 14, 693–699, 2020 (ref. 29); *J. Chem. Phys.* 155, 214304, 2021 (ref. 49).

Action taken. We have emphasized on page 9 (see lines 239-241, marked-up version) that the agreement with experiments is better under resonance conditions.

Reviewer's comment n.2: *“The Authors assume a fixed tip-molecule distance, how does the EET depend on that parameter?”*

Authors' answer. We thank the Reviewer for pointing out this aspect that we did not delve into before. For tip structures very similar to the ones used in our work, it has been previously shown in the case of photoluminescence TEPL imaging (*Nat. Photonics* 14, 693–699, 2020 (ref. 29); *J. Chem. Phys.* 155, 214304, 2021 (ref. 49)) that an increase of the tip-molecule separation leads to a broadening of the local field spatial distribution over the molecular space, as well as a decrease of its maximum intensity. Both conditions result in less-resolved images where sub-molecular features progressively vanish as the tip-molecule distance increases.

Inspired by the Reviewer comment we decided to look at the effect of tip-molecule distance on EET, too. New simulations at fixed D-A distance but for different tip-donor separations have been performed, revealing that overall RET_{eff} slightly increases by enlarging the tip-donor distance. This finding is in agreement with results illustrated in Fig. 6 where the absolute value of RET_{eff} increases as the system is moved further from resonance. In both cases, namely moving out of resonance or enlarging the tip-donor separation, the plasmonic system is screening the direct D-A coupling less efficiently and both donor plasmon-mediated radiative and non-radiative decays are smaller, thus favoring EET.

Action taken. We have performed additional simulations that can be found in the new SI section S.4, Fig.S3

Reviewer's comment n.3: *“How does the EET change when the tip addresses the other excited state (represented by the red arrow in Fig. 4a)”*

Authors' answer. We thank the Reviewer for raising this question. Based also on comment n. 5 of Reviewer 3 we have added the other donor state contribution to Eq. 6 as we realized that both donor degenerate states can get equally populated upon tunneling, see also lines 144-151 on p.5, marked-up version. The whole picture does not change: an exponential decay trend for the different plasmonic tip conditions is still observed. However, upon adding this other contribution we notice that the tip structure of Fig. 3a yields decay curves which almost perfectly match the experiments near the resonance condition (new Fig. 6).

At the shortest distance a small discrepancy is still observed, perhaps due to some effects going beyond classical electromagnetic interactions, which are anyway of minor relevance in the present case.

Action taken. New simulations that include the contribution of donor state S2, which was previously neglected, have been performed. The new results are included in the new Fig. 6.

Reviewer's comment n.4: *“Some assumptions regarding the molecules are not entirely correct. In the experiments reported in ref. 30, the excited state of PdPc lies slightly below the second excited state of H₂Pc, which should therefore not participate in the EET (as assumed by the model). However, one may include a “degeneracy” of the first excited state due to the tautomerization process, which occurs in such a configuration (see ref. 26). This issue should be discussed in the paper.”*

Authors' answer. We agree with the Reviewer that previous works have shown that tautomerization of H₂Pc can lead to experimentally observed features in the corresponding STML spectra. However, we believe that in our work this issue is not relevant. In Ref. 26 (pointed out by the Reviewer) the authors do experimentally observe tautomerization when the tunneling current pass through H₂Pc (current-induced tautomerization, see also *Science* 317,1203–1206, 2007 (new ref. 75)). In our case H₂Pc is the acceptor species, which is never directly excited by tunneling electrons, and backwards energy transfer is not directly probed as in Ref. 26. We also remark that in the experimental work of Cao et al. (ref.30) that we model here there is no evidence nor mention of H₂Pc tautomerization, which makes us think that it is not experimentally observed.

Action taken. We have added in the new “Computational Details” section (under “Methods”) , see lines 322-325 marked-up version, a mention to H₂Pc tautomerization, stating that it is not observed in the present case.

Reviewer's comment n.5: *“The model presented in this paper takes the excited state energies from the calculations, whereas the spectral overlap from the experiment. It is better when such parameters are consistent.”*

Authors' answer. We politely disagree with the Reviewer on the spectral overlap statement. Indeed, even if we wanted to compute it, it would require additional vibronic simulations (on top of TDDFT) which are typically rather involved and always require special care and efforts, thus clearly going beyond the scope of this work. The quantitative theoretical estimation of the spectral overlap is a well-known long-standing problem in the field of molecular energy transfer. Multiple times spectral overlaps have been taken from experiments, while excitation energies, which are more easily accessible through theory, have been obtained at ab-initio level (see for instance Cupellini, L, Corbella, M, Mennucci, B, Curutchet, C. *WIREs Comput Mol Sci.* 2019; 9:e1392.; Scholes GD. *Annu Rev Phys Chem.* 2003, 54,57–87 ; You ZQ, Hsu CP. *J Phys Chem A.* 2011, 115, 4092–4100). Furthermore, we point out that the main finding of the present work —understanding the role of the plasmonic system in mediating EET and proving that an exponential like decay of RET_{eff} can be reproduced even without resorting to a Dexter-like process—would not change if we had a different spectral overlap value as it appears just as scaling factor in eq. 9, which is the same for each D-A distance.

“Furthermore, TDDFT is known to not reproduce the excited state energies correctly and indeed there is around 0.2 eV mismatch between the theory and experiment.”

Authors' answer. We agree with the Reviewer that there is an error in the vertical excitation energies of around 0.2 eV, indeed within TDDFT such error is expected and acceptable.

However, we do not see how that small error could qualitatively change the whole picture, given that the absolute value of the excitation energy directly enters only in eq. 12, and a change of 0.2 eV would not lead to differences of order of magnitudes in the radiative emission rate.

Furthermore, many recent works on the same topic, see for instance Doppagne, B. et al. *Nat. nanotechnology* 15, 207–211, 2020 (ref. 28); Yang, B. et al. *Nat. Photonics* 14, 693–699, 2020 (ref. 29); Kong, F-F. et al. *Nat. Nanotechnol.* 17, 729–736, 2022 (ref. 31), do apply TDDFT, given its outstanding compromise between accuracy and computational cost.

In this view, and given that our new results almost perfectly match the experiments (see new Fig.6), we believe that an error of 0.2 eV is totally acceptable for the purpose of the present work.

Action taken. We have added in the section “Computational Details” (lines 320-321, marked-up version) a note to the systematic TDDFT error of 0.2 eV between simulated and experimental excitation energies.

Reviewer’s comment n.6: *“In the introduction to the model, the Authors refer essentially only to the works from their group. However, there exist several studies from other groups using similar approaches to address light-matter interaction between molecules and nanoscale plasmons and as of now the manuscript is not well placed in the context of the relevant literature. Some relevant works include:*

<https://doi.org/10.1021/acs.nanolett.7b05297> , <https://doi.org/10.1038/ncomms15225> , <https://doi.org/10.1103/PhysRevX.12.011012> , <https://doi.org/10.1038/s41467-019-11315-5> , <https://doi.org/10.1021/acs.jpcc.9b10256> and similar. Also, some of the cited experimental works (refs. 28, 29, 31) include theoretical calculations and should be discussed in that context. Ref. 31, in particular, features a theoretical description of EET in an STM environment.”

Authors’ answer. We thank the Reviewer for pointing out this lack of completeness from our side. We have added those references in the introduction, accompanied by a discussion on other theoretical models used to describe EET in an STM environment.

Action taken. We have added in the “Introduction” additional references and a discussion on other theoretical models to describe EET in STM environment (see lines 50-57 on p.2 and new refs.57-61)

Reviewer’s comment n.7: *“Ref. 31 reports on the observation of coherence in EET. Can the model presented in this work reproduce such behavior?”*

Authors’ answer. We thank the Reviewer for pointing out this curious aspect. In Ref. 31 the type of coherence that is detected is essentially related to exciton formation, which is something we can numerically assess within our model. In our case, the total D-A coupling mediated by the metal ($V_0 + V_{\text{met}}$, see eq.9) is ≈ 5 meV at the shortest D-A distance considered (1.59 nm), which clearly cannot lead to a significant exciton splitting, as detected in Ref. 31, where the predicted coupling is ≈ 5 times larger. We also point out that in Ref. 31 different D, A molecules are used and they are separated by 1.41 nm.

In agreement with our predicted small coupling value, in the experiment of Cao, S. et al. (ref. 30) coherence and exciton formation are indeed not observed nor mentioned.

Action taken. We have added in the “Computational Details” section (see lines 325-328) a mention to exciton formation and coherence for the present case.

Reviewer’s comment n.8: *“As the Authors describe, Eq. 10 includes also the Dexter-like energy transfer. Can the Authors quantitatively extract its contribution at short distances”*

Authors’ answer. This is something that can be quantitatively done by directly inspecting the density overlap contribution to eq. 10 assuming that it is not substantially modified by the tip structure, which is a reasonable assumption given that the shortest D-A distance is already rather large (1.59 nm) and that the tip effect on modifying the underneath D electron density is small for such tip structures and tip-molecule distances (see J. Chem. Phys. 155, 214304, 2021 (ref.49)). We numerically check both exchange and overlap terms of eq. 10 at the shortest distance and their contributions are respectively ≈ 4 and 5 orders of magnitude smaller than the full coupling V_0+V_{met} , thus corroborating that bare coulombic interactions dominate and Dexter-like energy transfer is of minor relevance here.

It is worth noting that in the supporting information of ref. 31 (SI section 5) the authors do observe a similar negligible contribution for the same terms responsible for a Dexter-like process. In that case their contribution is ≈ 6 orders of magnitude smaller than the total coupling value.

Action taken. We have emphasized on page 9 that the terms appearing in eq. 10 relating to a Dexter mechanism are many orders of magnitude smaller than the purely coulombic electromagnetic term, see lines 198-203 on p.9.

Minor comments:

1. *In the introduction, one reads “This approach leads to strong photoluminescence signals” – the referred approach (from the previous sentence) involves both photoluminescence and electroluminescence, so this sentence should be rephrased for clarity.*

Thank you for pointing this out, it has been corrected.

2. *Bottom of page 8 – there is a typo (“efficiency”).*

Thank you for spotting the typo, it has been corrected.

Reviewer #3:

We thank the Reviewer for their overall positive assessment: *“The paper is interesting potentially for the broader physical-chemistry community, mostly then for the community studying energy transfer on the nanoscale, and for the light-emission-in-STM community. It is also well written (up to some typos). The theory is mostly described clearly, although further clarifications are needed (see below)”* and *“... The paper is certainly publishable but I think that it requires an extension addressing the points below and potential concerns of my colleagues. That would make the work more complete and conclusive - especially because of its ambition to explain a particular experiment...”* .

We carefully address the Reviewer’s concerns below.

Reviewer's comment n.1: *“The work fits into the context of former work (I believe that more work, particularly theory from other groups, could be cited in this context)”*

Authors' answer. We thank the Reviewer for pointing out this aspect. We have added more references related to theoretical strategies adopted by other groups in the same context.

Action taken. We have added on lines 50-58 of “Introductions” more references and discussions on other modeling strategies. In particular new references 57-61 have been added on p.2 .

Reviewer's comment n.2: *“My main concern is that in the dimer configuration, the resulting coupling between the molecular excitons - particularly the "direct" vacuum contribution - is going to depend on the overlap of the orbitals of the donor and acceptor molecules and is going to be proportional to the exact exchange term (for concreteness, in the HOMO-LUMO approximation - which the authors correctly do not adopt - it would be $\sim \iint \psi_{\text{HOMO}_D}(\mathbf{r}) \psi_{\text{HOMO}_A}(\mathbf{r}) \psi_{\text{LUMO}_A}(\mathbf{r}') \psi_{\text{LUMO}_D}(\mathbf{r}') / |\mathbf{r}-\mathbf{r}'| d^3r d^3r'$). Clearly, this integral (and any other integral describing e.g. charge transfer) can only be evaluated correctly if the molecular orbitals of the monomers are correctly described at the position of the other molecule (more precisely, in the overlap region), i.e. far away from the nuclei of the respective monomers. But since gaussian basis is considered in the calculation, the asymptotic spatial dependence of the orbitals is actually described incorrectly as decaying with the gaussian tail. It is known that this issue leads to qualitative differences when the orbital density of states $|\psi(\mathbf{r})|^2$ is evaluated at larger distances from the molecule as done e.g. to simulate STM images. I therefore also doubt that the Dexter transfer, and particularly its dependence on the molecule-molecule separation, can be describe correctly in this model using this basis set.”*

Authors' answer. We agree with the Reviewer that gaussian basis sets are known not to faithfully reproduce the correct orbital spatial dependence far away from the nuclei of the monomers, but it is also true that enlarging the basis set size, especially with polarization and diffuse functions, is usually expected to improve such deficiency (see *J. Chem. Theory Comput.* 2010, 6, 3, 597–601; *J. Chem. Phys.* 106, 9639–9646, 1997; Stone, Anthony. *The theory of intermolecular forces.* oUP oxford, 2013.; Helgaker, Trygve, Poul Jorgensen, and Jeppe Olsen. *Molecular electronic-structure theory.* John Wiley & Sons, 2013.).

Inspired by the referee's comment we calculated the “direct” vacuum contribution enlarging the basis-set for the shortest D-A distance (1.6 nm), moving from 6-311G** to aug-cc-pVDZ or def2-TZVP. Even with a bigger basis set, the contribution to the total coupling due to exact exchange and direct density overlap (see eq.10) is respectively 4 and 5 orders of magnitude smaller than the purely electrostatic interaction (see also comment n.8 of Reviewer 2).

We also note that the same DFT functional and basis set have been used before to model D-A interaction in a similar STM junction (*Nat. Nanotechnol.* 17, 729–736, 2022 (ref. 31)).

Action taken. We have added on lines 298-301 (new “Computational Details” section) that increasing the basis set size does not lead to sizable differences in the direct overlap and exchange contributions of eq. 10 that still remain negligible, already at the shortest distance.

“ Another reason that makes me doubt this approach is the hybrid functional used by the autors which does not implement the exact exchange at larger distances (for that a range-separated functional or the exact HF would be needed). That is, I think that the present

approach cannot describe the molecule-molecule distance dependence of an important component of the energy transfer mechanism that may still play a significant role in the experiment of Cao et al.

Authors' answer. As replied above, previous works have shown that similar functionals and basis sets have been used for similar systems (see also Imada, H. et al. Nature 538, 364–367, 2016 (ref. 26)). Anyway, to fully deal with the issue raised by the Reviewer we additionally computed the direct vacuum contribution at the shortest D-A distance using CAM-B3LYP. Even in that case the overlap and exchange contribution to eq. 10 that should account for a Dexter-like process are ≈ 5 orders of magnitude smaller than the purely electrostatic term (1st term of eq.10), thus confirming that the latter is dominating.

Action taken. We have added on lines 298-301 (“Computational Details” section) that increasing the basis set size as well as using a range-separated functional does not lead to sizeable differences in the direct overlap and exchange contributions of eq.10 that still remain negligible compared to the purely classical electrostatic term.

“ I am also unsure about how the authors distinguish between the "electromagnetic" contribution and the "Dexter" mechanism. The Dexter mechanism is itself an effect of Coulomb interaction, and I believe that it is in principle included in the exchange part of eq. 10 (the authors may correct me and clarify this point). For the above mentioned reasons I think that, in the present form, the paper is not able to conclusively disentangle the mechanisms at play in the experiments of Cao et al. (perhaps this would explain the discrepancy with the experiment observed for small distances in Fig. 6 where the theory shows an incorrect trend showing a much weaker distance dependence than the experiment).”

Authors' answer. We relate the Dexter contribution to the exchange contribution of eq. 10. This term is many orders of magnitude smaller than the first one, which represents bare classical electromagnetic interactions and is always dominating, already at the shortest D-A distance (1.6nm). We point out that the same conclusion was drawn in Kong et al. Nat. Nanotechnol. 17, 729–736, 2022 (ref.31). In the new Fig.6, where we now include also the other donor state contribution (see comment n.5 of the same Reviewer, below) we show that simulations performed close to resonance conditions almost perfectly match the experiments, and only a very small discrepancy is observed at shortest distances, corroborating that in the present case a Dexter-like mechanism is of minor relevance, if any.

Action taken. We have performed all simulations again including the previously neglected contribution of the other donor state to RET_{eff} . The new results are now illustrated in new Fig. 6.

Reviewer's comment n.3: *“Having said that, it is true that the authors concentrate on the electrostatic interaction between the two molecular excitons. I agree that particularly the metal-mediated component could be described with reasonable accuracy even on the level of theory B3LYP/6-31G(d)**++. This part of the discussion is certainly interesting, including the study of the effect of plasmonic resonance on the energy transfer efficiency. However, I am missing a closer characterisation of the plasmonic properties of the tip-substrate system. What is the effective mode volume of this plasmonic "cavity"?”*

Authors' answer. We thank the Reviewer for recognizing the consistency of our computational setup, and for raising interesting points related to the geometrical features of the plasmonic system that are worth discussing. In the “Methods” section we briefly describe the PCM-NP modeling strategy that is used to treat the plasmonic system. There, we also show that in this classical response theory that is used to model the tip, the key quantity describing the frequency dependent response of the plasmonic system is the response matrix Q . In such complex object all plasmon modes are intrinsically embedded and considered, each one contributing more or less to the overall response depending on the external excitation frequency, so it is not easy to isolate single-mode contributions and corresponding effective mode volumes, as all modes are included in the full response that we simulate.

“What is the field distribution characterising the mode that is resonant with the excitons? How does it compare to other work? Also, when inspecting the geometry, I noticed that in Fig. 3b the atomistic protrusion is connected to the main body of the tip abruptly, creating thus a sharp corner (aparent in the zoom). Is this a reasonable geometrical representation? Is the sharp edge between the protrusion and the smoother bigger part of the tip influencing the plasmonic mode volume or perhaps even creating an unrealistic field distribution or resonance in the corner region? How does it influence the results? Similarly, the substrate is modelled as a disk of finite size. Is the result dependent on it's size? I think that at least these points deserve more attention of the authors and should be discussed and displayed in the paper (perhaps at least in the supplement). I think that this discussion is particularly important for this paper as it is targeting (semi) quantitative description of the system and directly compares the theoretical data with a concrete experiment.”

Authors' answer. In previous works (J. Chem. Phys. 155, 214304, 2021 (ref. 49); Nat. nanotechnology 15, 207–211, 2020 (ref. 28)) a detailed characterization of the picocavity field distribution of the same tip structures that we here use has been reported, as well as an extensive analysis of how those geometrical features of the tip protrusion can impact on the local field spatial distribution. Other works (Nat. Photonics 14, 693–699,2020 (ref. 29)); Nat. Nanotechnology 15, 207–211, 2020 (ref. 28)) on very similar systems have also illustrated that plasmon modes that do feature charge localization at the protrusion apex are those that mostly couple to the molecule. In all cases, despite the crude modeling of such complex tip structures with those sharp protrusions, it has been extensively proven that this kind of modeling can quantitatively match state of the art experiments on TEPL or STML. We kindly refer the Reviewer to those works where an extensive analysis of all these aspects have been thoroughly investigated.

Furthermore, in the new Fig. 6 we directly compare EET results for the two different tip structures of Fig. 3 (main text), quantitatively showing that different tip morphologies and disk shapes and sizes can indeed have a significant impact on the observed signal, thus being perfectly in agreement with previous findings showing that the local field distribution on the molecules position is sizably affected by tip geometrical features. Unfortunately, no experimental characterization of the geometrical features of the plasmonic system is available in our case, but we note that one of the chosen computational setup does provide results in close agreement with experiments.

Action taken. We have added in the SI new section S.3, lines 40-44, a mention to the importance of tip geometrical features in changing the local field spatial distribution at the molecule's position together with some references where that effect has been carefully investigated. Furthermore, in lines 237-254 (main text) the importance of plasmonic geometrical features in affecting EET has been further remarked.

Reviewer's comment n.4: *"I am also wondering about the ability of the model - using a local description of the metal optical response - to properly capture the decay of the excitons into the electron-hole pairs and plasmons of higher k vectors of the substrate. Can the authors estimate how much the non-local effects in the metal could influence the lifetime of the monomer excitons? I believe that this could also rescale the results presented by the authors, although I understand that such a study is likely beyond the scope of the paper. Could the authors discuss this effect and potential limitations of their model in this respect?"*

Authors' answer. It is true that the present model does not account for nonlocality in the metal optical response. Nevertheless, many works investigating the coupling of molecular excitons with STM tips (Phys. Rev. X. 12, 011012, 2022 (new ref. 59); Nat. Photonics 14, 693–699,2020 (ref. 29); Nat. nanotechnology 15, 207–211, 2020 (ref. 28); J. Chem. Phys. 155, 214304,2021 (ref. 49)) with similar local descriptions of the metallic body have shown that such simulations can quantitatively match state of the art experiments. For instance, in ref. 29 Nat. Photonics 14, 693–699,2020 the predicted exciton lifetime in the picocavity (related to linewidth broadening) is actually underestimated, albeit slightly, with respect to the experimental data, pointing out that nonlocality should play a minor role.

Action taken. We have highlighted at lines 292-293 (" Computational Details" section, under "Methods") that nonlocality is not included in the present model.

Reviewer's comment n.5: *"Getting into more detail, I would like to ask the authors about the following statement: "Following a previous work [28], the donor state mainly excited by the tip due to tunneling electrons corresponds to the one whose dipole points along the molecule's aromatic lobe beneath the tip, illustrated schematically in Fig. 4a." Are the authors sure that this is a matter of excitation probability? It is the photon emission probability that is usually influenced by the tip position. If the molecule is excited by sequential electron tunneling, it is the density of states of the relevant molecular orbitals at the tip position that determines the excitation probability, not the orientation of the excitonic transition dipoles. And in this case HOMO's DOS is totally symmetrical (assuming that it is HOMO that is involved in the tip-mediated tunneling). LUMO's DOS's symmetry is reduced, but not substantially to preferentially excite just one of the two excitons (so even if LUMO is the orbital involved in tip-mediated tunneling the author's statement is not fully justified). Perhaps the authors should comment on this and probably modify the claim."*

Authors' answer. We thank the Reviewer for pointing out this aspect and we undoubtedly agree with their comment. That concept was wrongly expressed and applied by us. It is indeed the excited state whose transition density is mainly located near the tip protrusion that most strongly interacts with the plasmon and so can experience a significantly different emission probability. Indeed, the contribution to I_D (eq. 4) coming from the other excited state of the donor (whose transition dipole is depicted in red in Fig. 4a) is numerically negligible compared to the one originating from the state whose transition density strongly interacts with the plasmonic tip (represented by the blue transition dipole in Fig. 4a). Nevertheless, its contribution to I_A upon EET to the acceptor turns out not to be negligible. For this reason we have performed all calculations again including also the contribution of the other donor excited state to both I_D and I_A . Notably, all decay curves still feature an exponential decay trend, but now the tip structure of Fig. 3a, gives results that are much closer to the experimental data.

With regard to the excitation probability, based on previous findings (*Nat. nanotechnology* 15, 207–211, 2020 (ref. 28); *Nano letters* 19, 2803–2811, 2019 (new ref. 67)) and given that the bias voltage applied in the experiments is negative, we now assume that the tip removes one electron from the HOMO donor orbital and subsequent electron injection from the substrate takes place. This electron can be equally injected either in LUMO or LUMO+1 since these two orbitals are degenerate and there is no sound reason why the substrate should prefer one over the other (they are equally distributed over the substrate surface). Given that, both donor S1 and S2 degenerate excited states, which are respectively described mainly by HOMO->LUMO and HOMO->LUMO+1 transitions, are equally populated and so can contribute to the total RET_{eff} which is now reported in the new Fig. 6 and it is computed according to eq. 6 summing over the i index too (see also new eq. 15).

We thank the Reviewer for correctly noticing this subtle aspect.

Action taken. We have corrected that paragraph and re-performed all calculations including the other donor state contribution, see lines 144-152, marked-up version and new Fig. 6.

Reviewer's comment n.6: *“Also, in case of H2Pc the authors call the excited states quasi-degenerate. There is a separation of ~100 meV between the H2Pc excited states and the excitonic photon emission peaks (commonly denoted as Qx and Qy) can be clearly spectrally separated and identified in STM-based experiments. Is it still reasonable to talk about (quasi)degeneracy in this case? How would it modify the results if the states were fully treated as non-degenerate?”*

Authors' answer. We agree with the Reviewer that using the word “quasi-degenerate” here is not appropriate, we have corrected that improper wording. Anyway, the results would have not changed in any case, as even in the present case each state of H₂Pc enters into the modeling strategy by its own excited state properties (excitation energy, transition density, etc.) as obtained by the ab-initio TDDFT calculation. Practically speaking no assumption about quasi-degeneracy was made, it was just an inaccurate wording that we used to describe the small difference between excitation energies.

Action taken. We have corrected that paragraph, see lines 302-303 (“Computational Details” section under “Methods”).

Reviewer's comment n.7: *“Also, it is well known that the molecule of H2Pc features tautomerization when excited. Is this relevant for the present study? Could the authors discuss this point?”*

Authors' answer. We agree with the Reviewer that previous studies have illustrated the possibility of observing H₂Pc tautomerization in STM junctions. However (see also comment 4 of Reviewer n.2) they showed that tautomerization is usually observed when the STM-current passes directly through the H₂Pc molecule (*Science* 317,1203–1206, 2007 (new ref. 75)), which is not the case here. Moreover, in the work of Cao, S. et al that we here target, there is no experimental evidence of such phenomenon. For these reasons we believe that such a process is not relevant in the present case.

Action taken. We have mentioned the issue of tautomerization at lines 322-325 (“Computational Details” section, under “Methods”), saying that it should be of no relevance in the present case.

Reviewer’s comment n.8: “*The authors implement the spectral overlap factor J (eq. 9) as a constant taken from [30]. Why don't they use the theory tools in hand to estimate it computationally to verify the value?*”

Authors’ answer. As answered to a similar comment of Reviewer 2. (comment n.5) calculations of spectral overlap are usually much more difficult and involved than getting excitation energies and corresponding transition densities. It typically requires many assumptions on the vibronic model being used and often it cannot capture the overlap accurately enough in the realistic systems being probed. Besides, previous works have shown that errors in excitation energies can lead to large differences in the predicted spectral overlaps and so corresponding EET rates (Cupellini, L, Corbella, M, Mennucci, B, Curutchet, C. *WIREs Comput Mol Sci.* 2019; 9:e1392.; Scholes GD. *Annu Rev Phys Chem.* 2003;54:57–87 ; You ZQ, Hsu CP. *J Phys Chem A.* 2011,115:4092–4100). Moreover, the main finding of the present work —understanding the role of the plasmonic system in mediating EET and proving that an exponential like decay of RET_{eff} can be reproduced even without resorting to a Dexter-like process—would not change if we had a different spectral overlap value as it would be constantly the same value for each D-A distance (see eq.9).

On these grounds we believe that an accurate theoretical evaluation of the spectral overlap is beyond the scope of the present work.

Reviewer’s comment n.9: “*Related to the spectral overlap, could the authors comment on the effect of energy shifts of the excitonic peaks induced by the plasmonic environment (plasmonic Lamb shift) and the static Stark shift due to the tip-sample voltage (see [Roslawska et al.PRX 12, 011012 (2022)])? “*”

Authors’ answer. As shown in the work of Roslawska et al., and in agreement with previous findings e.g. e.g. *Nat. Photonics* 14, 693–699,2020 (ref. 29); *J. Chem. Phys.* 155, 214304,202 (ref. 49), static Stark shift and plasmon-induced Lamb shift typically results in very small shifts (few meV) of excitation energies of molecules placed very close to such tip structures. In this sense, speaking of the capability of such models to quantitatively match the experiments, this little contribution can be often neglected considering that there are other more important missing experimental details (such as a proper characterization of the tip structure and protrusion geometry) that can have much larger effects on the overall response. Furthermore, given that our calculations reported in Fig. 6 are done for different D-A distances where only the acceptor is moved -the donor is always fixed underneath the metallic tip- even if there was a sizeable shift, that shift would likely involve the donor molecule most significantly and not the acceptor, as the latter is always farther from the tip. Therefore, this hypothetical contribution would always be the same for each separation since the donor position is always fixed, so spectral overlap will not change for different distances based on these effects.

With our model we can have access to the plasmon-induced Lamb shift, which is related to the real part of the molecule plasmon self-interaction (see eq.13 and ref.49) and for our tip-donor setup it is ≈ 8 meV when the tip response is on resonance with the D transition (using either tip structure of Fig. 3), thus confirming that this contribution would be negligible even under the strongest-coupling condition.

Action taken. We have reported the computed Lamb-shift value under the strongest-coupling condition at lines 315-317, marked-up version.

“Also, the direct and metal-mediated exciton-exciton coupling between the neighbouring monomers could cause spectral shifts, is this taken into account? The shifts could change the spectral overlap between the donor and the acceptor as a function of the relative position of the molecules with respect to the tip and the molecule-molecule separation. Is this effect potentially important or can it be neglected? Could it be quantified?”

Authors’ answer. At the shortest D-A distance, the total exciton-exciton coupling evaluated as V_0+V_{met} (see eqs.9-11) is ≈ 5 meV thus clearly showing that it is not large enough to provide sizeable shifts in the excitation energies and corresponding spectral overlap. This finding is perfectly in agreement with the experiment being modeled of Cao, S. et al. where there is no mention of a distance dependent shift in the STML emission energies of both donor and acceptor as well as changes in the estimated spectral overlap for different separations.

Reviewer’s comment n.10: *“I am also wondering about the modification of the ground state when DFT calculations are performed in the presence of the substrate and the tip. Do the authors consider this in their model, or is the presence of metal only modifying the linear-response time-dependent part of the calculation? It would be interesting to see how e.g. the image-charge potential of the metal modifies the electronic structure of the molecules. Is the model of the authors able to describe this? I understand that this point (if not implemented already) is likely beyond the scope of the present paper, I am just asking out of curiosity. “*

Authors’ answer. We thank the Reviewer for raising this point. Indeed, as correctly guessed, the ground state density is modified by the presence of the nearby plasmonic system and this effect is already taken into account in the presented theoretical model. However, in agreement with previous findings (see Fig.9 of J. Chem. Phys. 155, 214304,2021 (ref. 49)) this effect is basically negligible for the setup being investigated.

Action taken. We have added a sentence on page 12, lines 307-309 highlighting this aspect which was already included in the presented results but not sufficiently well-expressed in the text.

Reviewers' comments:

Reviewer #1 (Remarks to the Author):

The authors' reply and their revised manuscript satisfactorily address my questions. Thus, I recommend the manuscript to be accepted for publication in Communications Chemistry.

Reviewer #2 (Remarks to the Author):

The Authors have submitted a revised manuscript where they clarify the issues raised by the referees and overall improve the quality of the paper. The main result of the work (Fig. 6) now reproduces the previously reported experiments more accurately after including other relevant excited states. The discussion is more extended and highlights the key role of the tip in the process. Overall, I am happy to recommend the manuscript for publication once some minor comments are implemented.

1. The Authors consider that two excited states of the acceptor molecule participate in the EET, which is not accurate with respect to the experiment they analyze. For the H2Pc molecule the energy of the S2 state (often labeled as Qy, experiment: 1.93 eV) is higher than the energy of the donor excited state (experiment: 1.92 eV) therefore the statement in the line 319 is not correct (“donor (S1/S2 = 2.11/2.11 eV) and acceptor molecules (S1/S2 = 2.02/2.05 eV)”). In this work, the acceptor states should be rather labeled as S1 and S1' (or similar) because, in contrary to what the Authors write, the molecule will tautomerize during such a measurement (even if not discussed in ref. 30). In the case of work from ref. 30, the tunneling current (excitation) is relatively high and the integration time of a spectrum is long enough that the molecule is expected to switch many times and the two tautomer configurations (and their contributions to the detected intensity) will average in time. An example where this is not the case is shown in Fig. 4 of ref. 26, where the measurements are done at lower currents and the tautomerization is a much less frequent event resulting in an intensity oscillation from spectrum to spectrum. As reported in the ref. 28, the excited states of the two tautomers can have different energies, which could account for the values proposed in this paper (2.02/2.05 eV). Also, no current signal (as in ref. 75) is expected in such a measurement since the tip is located on PdPc. Therefore, my suggestion is to rewrite the part in lines 317-328 accounting for the two tautomer configurations, that is S1 state of the tautomer #1 and the S1 state (S1') of the tautomer #2, which can be different in energy, justifying the choice of 2.02/2.05 eV values. Here, what the Authors should verify is whether the fact that at a given instance in time only one of the two tautomers is available will change the final energy transfer efficiency.

2. Line 325,327: “exciton formation” – I find this description ambiguous as it is a common practice to say that once the molecule is in the excited state an exciton is formed. I assume the Authors refer here to an excitation that is delocalized over more than one molecule and therefore should be more specific in the wording.

Reviewer #3 (Remarks to the Author):

The authors have satisfactorily answered the questions and comments of reviewers and I can therefore recommend the paper for publication.

Reviewer #1:

We thank the Reviewer for their useful comments and suggestions thanks to which the manuscript has undoubtedly improved. We are pleased to see that the manuscript is now recommended for publication.

Reviewer #2:

We thank the Reviewer for acknowledging that *“The authors have submitted a revised manuscript where they clarify the issues raised by the referees and overall improve the quality of the paper. The main result of the work (Fig. 6) now reproduces the previously reported experiments more accurately...”* and for recommending our manuscript for publication after implementing few remaining minor comments. We address these points below.

Reviewer’s comment n.1: *“The Authors consider that two excited states of the acceptor molecule participate in the EET, which is not accurate with respect to the experiment they analyze. For the H₂Pc molecule the energy of the S₂ state (often labeled as Q_y, experiment: 1.93 eV) is higher than the energy of the donor excited state (experiment: 1.92 eV) therefore the statement in the line 319 is not correct (“donor (S₁/S₂ = 2.11/2.11 eV) and acceptor molecules (S₁/S₂ = 2.02/2.05 eV”). ”*

Authors’ answer. We thank the reviewer for raising this issue. By reading the section “Estimation of the spectral overlaps for the D-A dimers” (in “Methods”) of the article of Cao et al. (ref. 30) we can only understand that the Q_y band intensity is neglected when the spectral overlap between PdPc and H₂Pc is estimated. The justification that is given is that its energy is reported to be 0.01eV higher than the PdPc energy level. Nevertheless, this does not necessarily mean that EET to Q_y is not taking place. Indeed, it is the actual spectral overlap value, which depends on the vibronic structure of the states considered, that properly accounts for energy conservation and thus the possibility of experiencing EET between two states.

Considering that the estimation of the spectral overlap between Q_{PdPc} and Q_y is experimentally missing and that 0.01eV is a very small energy difference, we cannot rule out the possibility of having EET from PdPc to Q_y. Furthermore, in the experiments of Cao et al. when the tip is located on H₂Pc a weak emission signal in the Q_{PdPc} frequency range is still observed (see reddish shades on H₂Pc of Fig.3d), which could corroborate the possibility of having EET from PdPc to the Q_y band of H₂Pc.

Given that the experimental value of the spectral overlap between Q_{PdPc} and Q_y is missing, in the revised version different values have been tested. The results reported in the new Supplementary Figure 4 show that the exponential decay behavior of RET_{eff} is consistently observed even if the spectral overlap value between Q_{PdPc} and Q_y is sizably changed, thus confirming the robustness of our findings.

Action taken. We have emphasized at lines 322-327 that we include both acceptor states in EET and that the main conclusions drawn in our work are not affected by the adopted value of the spectral overlap between Q_{PdPc} and Q_y bands. Supplementary Figure 4 has been added to show this.

“ In this work, the acceptor states should be rather labeled as S₁ and S₁’ (or similar) because, in contrary to what the Authors write, the molecule will tautomerize during such a measurement (even if not discussed in ref. 30). In the case of work from ref. 30, the tunneling current (excitation) is relatively high and the integration time of a spectrum is long enough

that the molecule is expected to switch many times and the two tautomer configurations (and their contributions to the detected intensity) will average in time. An example where this is not the case is shown in Fig. 4 of ref. 26, where the measurements are done at lower currents and the tautomerization is a much less frequent event resulting in an intensity oscillation from spectrum to spectrum. As reported in the ref. 28, the excited states of the two tautomers can have different energies, which could account for the values proposed in this paper (2.02/2.05 eV). Also, no current signal (as in ref. 75) is expected in such a measurement since the tip is located on PdPc. Therefore, my suggestion is to rewrite the part in lines 317-328 accounting for the two tautomer configurations, that is S1 state of the tautomer #1 and the S1' state (S1') of the tautomer #2, which can be different in energy, justifying the choice of 2.02/2.05 eV values. Here, what the Authors should verify is whether the fact that at a given instance in time only one of the two tautomers is available will change the final energy transfer efficiency. "

Authors' answer. We thank the Reviewer for carefully pointing our attention to the tautomerization issue. Indeed, previous works showed that tautomerization occurs through the H₂Pc excited states (ref. 28). Therefore, we agree with the Reviewer that tautomerization may continuously take place in the experiments of Cao et al., even if it is not mentioned, given the experimental tunnelling conditions and previous findings of ref. 26.

In our modelling, we have not explicitly dealt with tautomerization, but we have included both H₂Pc excited states. The latter aspect turned out to be an effective way of including tautomerization, which is therefore already present in the results we show, although approximately. Indeed, according to TDDFT the acceptor excited states S₁/S₂ are close in energy (≈ 30 meV energy difference) and feature two mutually orthogonal transition dipoles that have numerically the same magnitude. On the other hand, upon tautomerization, the Q_x transition dipole of one tautomer will change its orientation by 90° but its magnitude and Q_x energy will not change much, as shown in ref.28. Therefore, by including both S₁ and S₂ states of H₂Pc, as we are currently doing, we are already including in an effective way the average effect of the two tautomers since both S₁ and S₂ states are just ≈ 30 meV apart in energy and their transition dipoles have same magnitude and are mutually orthogonal, exactly as the S1 and S1' states mentioned by the Reviewer.

Based on these arguments, we believe that a more through description of the two tautomers will not change the main conclusions of our work, and its detailed inclusion would thus go beyond the scope of the present study.

The discussion on tautomerization at lines 328-334 has been modified accordingly.

Action taken. We improved the discussion on the tautomerization issue at lines 328-334.

Reviewer's comment n.2: *"Line 325,327: "exciton formation" – I find this description ambiguous as it is a common practice to say that once the molecule is in the excited state an exciton is formed. I assume the Authors refer here to an excitation that is delocalized over more than one molecule and therefore should be more specific in the wording."*

Authors' answer. Yes, what the Reviewer assumes is correct, we indeed were referring to an excitation delocalized over more than one molecule. We agree that the wording could generate confusion. We have edited that paragraph to make it clearer what we mean. Thanks.

Action taken. We have modified that paragraph which now corresponds to lines 335-338, marked-up version.

Reviewer #3:

We thank the Reviewer for their thorough comments and suggestions thanks to which the manuscript has certainly improved. We are glad to see that the manuscript is now recommended for publication.

REVIEWERS' COMMENTS:

Reviewer #2 (Remarks to the Author):

The Authors have addressed all the comments, added more data and explanations. I am happy to recommend that work for publication.